# LOB-Bench: Benchmarking Generative AI for Finance – an Application to Limit Order Book Data

Peer Nagy [* 1 2]   Sascha Frey [* 3]   Kang Li [4]   Bidipta Sarkar [2]   Svitlana Vyetrenko [5]   Stefan Zohren [1]
Anisoara Calinescu [3]   Jakob Foerster [2]

## Abstract

While financial data presents one of the most challenging and interesting sequence modelling tasks due to high noise, heavy tails, and strategic interactions, progress in this area has been hindered by the lack of consensus on quantitative evaluation paradigms. To address this, we present **LOB-Bench**, a benchmark, implemented in python, designed to evaluate the quality and realism of generative message-by-order data for limit order books (LOB) in the LOBSTER format. Our framework measures distributional differences in conditional and unconditional statistics between generated and real LOB data, supporting flexible multivariate statistical evaluation. The benchmark also includes commonly used LOB statistics such as spread, order book volumes, order imbalance, and message inter-arrival times, along with scores from a trained discriminator network. Lastly, LOB-Bench contains "market impact metrics", i.e. the cross-correlations and price response functions for specific events in the data. We benchmark generative autoregressive state-space models, a (C)GAN, as well as a parametric LOB model, and find that the autoregressive GenAI approach beats traditional model classes. Code and generated data are available at: https://lobbench.github.io/.

## 1. Introduction

Practitioners have long been interested in high-quality synthetic financial data, which is especially difficult in the high-frequency domain due to high noise, heavy tails, and multi-agent interactions. Generative AI (GenAI) is currently revolutionizing different fields, ranging from natural language processing to image generation and real world applications. Perhaps surprisingly, the backbone of these methods is simply self-supervised pre-training on large datasets using a next-token prediction loss on autoregressive sequence models (Nie et al., 2024; Dubey & et. al., 2024; Liu et al., 2024).

Recently, Nagy et al. (2023) applied this paradigm to *limit order books* (LOB), the mechanism through which stock markets keep track of buy and sell orders to determine any-time prices. Specifically, in contrast to prior works, which model only high level features (Cont et al., 2010; Coletta et al., 2022; Byrd et al., 2020), this approach learns a *token-level* distribution over messages in the LOBSTER dataset (Huang & Polak, 2011).

An *accurate*, *low level* generative model of the financial system is extremely valuable from a societal and commercial point of view. For example, it could unlock better mechanism design, stability analysis, or learned-algorithms (e.g. order execution (Frey et al., 2023)) by providing counterfactuals.

A key question then is how to determine the realism and trustworthiness of GenAI, and of other generative financial models. On the one hand, for high-level approaches and "old school" agent-based modelling (Byrd et al., 2020; Chiarella & Iori, 2002; Paulin, 2019; Llacay & Peffer, 2018) the evaluation is usually based on a qualitative analysis of

*Equal contribution [1]Oxford-Man Institute of Quantitative Finance, University of Oxford [2]Foerster Lab for AI Research, University of Oxford [3]Department of Computer Science, University of Oxford [4]Department of Statistics, University of Oxford [5]J.P. Morgan AI Research. Correspondence to: Peer Nagy <peer.nagy@eng.ox.ac.uk>.

*Proceedings of the 42nd International Conference on Machine Learning*, Vancouver, Canada. PMLR 267, 2025. Copyright 2025 by the author(s).

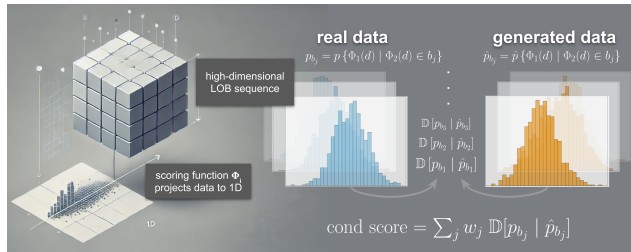

Figure 1: **Schematic of LOB-Bench methodology** for conditional distributional evaluation

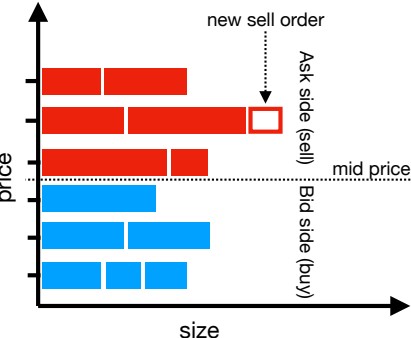

Figure 2: **Schematic of the LOB.** Not immediately executable orders are placed in First-In, First-Out (FIFO) queues at the specified limit price. Sell limit orders are placed on the Ask side (red) and Buy limit orders on the Bid side (blue).

whether the model reproduces known high-level patterns – "stylized facts" from the literature, "impact" or the famous "square-root law" (Tóth et al., 2016; Brokmann et al., 2015; Almgren et al., 2005b). However, most of these metrics are unquantifiable and may be disconnected from ground-truth data.

On the other hand, for GenAI the standard evaluation for pre-training is simply *cross-entropy*, i.e. how closely the model is able to predict the next token on held-out data. Unfortunately, this does not capture how the model performs under autoregressive sampling, when *generating* sequences of data one token at a time, where error accumulation can cause distribution shifts. In many applications of GenAI this is not a problem, since the pre-trained models are merely used as *starting points* for task specific finetuning (e.g. RLHF), rather than in their "bare" form. In contrast, we want to evaluate the pre-trained models in the *sampling* regime to unlock the mentioned use-cases.

To address this, we propose a general framework for evaluating the similarity between the distribution induced by generative LOB models and the ground-truth data. At a high level, our *unconditional* evaluation consists of three steps. We first introduce a set of *aggregator functions*, $\Phi$, which map from high-dimensional time series LOB data into a set of 1d subspaces. Secondly, we compute histograms to estimate distributions for the ground-truth and generated data in these subspaces and, finally, use a distance metric, e.g. $L_1$, to compare differences in these estimates. Some of the aggregator functions chosen are closely inspired by metrics used in literature (Vyetrenko et al., 2021; Paulin, 2019; Chiarella & Iori, 2002; Cont, 2001). They also directly relate to *generative adversarial networks* (Goodfellow et al., 2014), where the discriminator network is equivalent to a *worst-case* aggregator function for a given generator.

For *conditional* distributional evaluation, we first apply an aggregator function and group these results into "buckets" based on the conditioning variable. We then score each of the resulting conditional distributions using the process described earlier. This approach enables, for example, assessing whether the distribution of bid-ask spreads, conditioned on the time of day, aligns with the corresponding conditional distribution in real data. To derive a single metric, we compute the average loss across the conditioning buckets, weighted by the probability of each bucket. Furthermore, we can also use this to evaluate model-drift by aggregating on the *sampling step* and comparing to the unconditional data, which is a good proxy for model-derailment in open-loop sampling. See Figure 1 for a process schematic.

We test our evaluation framework on five different generative models: four modern GenAI models (Coletta et al., 2022; Nagy et al., 2023; Peng et al., 2023; 2024) and a widely-used classic model as a baseline (Cont et al., 2010). All models are tested on data of Alphabet Inc (GOOG) and Intel Corporation (INTC) stock. We don't present detailed results for the *Coletta* model trained on INTC, because the architecture was developed only for small-tick stocks and therefore fails on INTC data (Coletta et al., 2022). We find evidence of "model derailment," since the distance scores increase for longer unrolls (Figure 5). We also find that the *LOBS5* model is best able to reproduce the standard *price-impact curves* that are well-known in the economics and finance literature (Eisler et al., 2012).

Our contributions are summarized as follows:

**A novel LOB benchmark for distributional evaluation:** the first LOB benchmark focused on full distributional quantification of model performance. This addresses limitations of prior work, which relied on qualitative comparisons of stylized facts, making rigorous model comparisons infeasible and hindering research progress.

**Interpretable scoring functions for targeted improvements:** using intuitive scoring functions enables targeted model development and refinement.

**Difficult challenge of discriminator scores:** discriminator-based scoring sets a high bar for future generative models, even when most other statistics are closely aligned.

**Identification of a common failure mode:** divergence metrics, computed as distributional errors as a function of unroll step, highlight a prevalent failure mode to guide research.

**Ease of use and accessibility:** open-source, straightforward to apply benchmark which only requires data in the LOBSTER format.

**Extensibility** to additional scoring functions.

**Transferability to other domains:** the theoretical framework is adaptable to other high-dimensional generative time series tasks beyond LOB data.

We hope our benchmark will provide a much-needed starting point for evaluating GenAI models in finance and allow more machine learning scientists to develop new sequence models for this important and challenging domain. Our code and additional resources are available at: https://lobbench.github.io/.

## 2. Background

### 2.1. Limit Order Book (LOB)

Later sections of this paper rely on the reader's understanding of the mechanisms of electronic markets, so we briefly review them here. Public exchanges such as NASDAQ and NYSE facilitate the buying and selling of assets by accepting and satisfying buy and sell orders from multiple market participants. The exchange maintains an order book data structure for each asset traded. The LOB represents a snapshot of the supply and demand for the asset at a given time. It is an electronic record of all the outstanding buy and sell limit orders organized by price levels. A matching engine, commonly using a *price-time* priority mechanism, is used to pair incoming buy and sell order interest as mentioned in Bouchaud et al. (2018). Order types are further distinguished between limit orders and market orders. A limit order (Figure 2) specifies a price that should not be exceeded in the case of a buy order (bid), or should not be gone below in the case of a sell order (ask). A limit order queues a resting order in the LOB at the corresponding side of the book. A market order indicates that the trader is willing to accept the best price available immediately.

In real-time trading, injecting orders into the market induces other market participant activity that typically drives prices away from the agent. This activity is known as market impact (Almgren & Chriss, 1999; Almgren et al., 2005a). Presence of market impact in real time implies that a realistic trading strategy simulation should include deviation from historical data. Therefore, realistic market impact emulation is an important consideration in limit order book modelling.

### 2.2. LOB Models

LOB simulation is an important technique for evaluating trading strategies and testing counterfactual market scenarios. The extent to which results from such simulations can be trusted depends on how accurately they emulate real world environments. Traditionally, it is common to use historical market data to train and backtest a trading strategy, thereby making the assumption of negligible market impact. This is based on the premise of small agent orders and a sufficient time between consecutive trades (Spooner et al., 2018). However, the "no market impact" assumption is not valid for larger order sizes or a high frequency of orders. Agent-based methods naturally allow to study such

phenomena, which emerge as a consequence of multiple participant interactions, which are difficult to model otherwise. However, they are notoriously challenging to calibrate (Vyetrenko et al., 2021; Paulin, 2019). To circumvent calibration, conditional generative adversarial networks were used to learn simulators from historical LOB data, that are both realistic and responsive (Coletta et al., 2023). Most recently, an end-to-end autoregressive generative model that produces tokenized LOB messages in the spirit of generative AI was shown to achieve a high degree of realism (Nagy et al., 2023).

### 2.3. Autoregressive LOB models

In machine learning, autoregressive modelling is a key component of language models like GPT. By learning the probability distribution of the next token given the previous tokens, autoregressive language models can generate coherent text (Radford et al., 2019). Cross-entropy is a loss function commonly used to train classification models in deep learning. It measures the dissimilarity between the predicted class probabilities and the true class labels (Goodfellow et al., 2016). Cross-entropy loss is the negative log likelihood of the true class labels under the predicted distribution. Minimizing the cross-entropy is equivalent to maximizing the likelihood of the data (Murphy, 2012). The cross-entropy loss over a sample of size $N$, with $V$ classes can be expressed as:

$$L = -\frac{1}{N} \sum_{i=1}^{N} \sum_{v=1}^{V} y_i^{(v)} \log \hat{y}_i^{(v)},$$

where $y_i^{(v)}$ equals 1 if the true class is $v$ and 0 otherwise, and $\hat{y}_i^{(v)}$ is the predicted probability for class $v$. Cross-entropy loss heavily penalizes confident misclassifications and incentivizes the model to output calibrated probabilities that match the empirical distribution of the classes. Although it is different from the KL divergence, cross-entropy can be expressed as the sum of the entropy of the true distribution and the KL divergence between the true and predicted distributions (Cover & Thomas, 1999).

## 3. Related Literature

The LOB plays a crucial role in modern financial markets. With the FI-2010 dataset, Ntakaris et al. (2018) released the first publicly available high-frequency LOB dataset for benchmarking mid-price prediction models. This preprocessed dataset containst orders for five stocks on the Nasdaq Nordic market for ten consecutive days. Although useful and effective for preliminary tests and comparisons of LOB algorithms, FI-2010 does not allow a comprehensive evaluation of robustness and generalisation ability (Zhang et al., 2019). A similar benchmark for average price and

volume prediction in Chinese stock markets is provided by Huang et al. (2021). As with other currently available benchmarks, this work falls short of evaluating GenAI models with a fully distributional lens. Cao et al. (2022) propose a benchmark dataset, which plays a complementary role to LOB-Bench. With DSLOB, they provide a synthetic LOB dataset, generated by a multi-agent simulation with shocks, which generates labelled in- and out-of-distributions samples. In contrast, LOB-Bench does not require training on a specific dataset, and instead focuses on general-purpose model evaluation and comparison.

To evaluate the performance of generative models in the LOB environment, several studies have proposed relevant metrics. Coletta et al. (2023) investigated the interpretability, challenges, and robustness of conditional generative models. They grouped LOB states based on certain attributes and statistics and then performed conditional generation on these groups. Vyetrenko et al. (2021) proposed several statistics to assess the realism of LOB simulators, such as order arrival rate, order distance distribution, and price volatility, whilst Paulin (2019) further considers lagged autocorrelations, and liquidity of trades.

In summary, although some studies have addressed the evaluation of generative models for LOBs, a unified benchmarking framework is still lacking. Existing research often uses *qualitative* methods to compare statistical regularities of generated data with real data, lacking quantitative evaluation metrics. Therefore, establishing a comprehensive benchmarking framework for evaluating LOB generative models is essential for advancing the field.

## 4. Evaluation Framework

As the success of LLMs has shown, generative models can already achieve impressive performance by autoregressive training, or "next-token prediction" alone. However, not all model classes are auto-regressive or allow the explicit computation of conditional "next-token probabilities," prohibiting cross-entropy based evaluation or calculating model perplexity (Chen et al., 1998). However, there is still a need to evaluate such model classes, where we can merely sample data. Another reason why single-token cross-entropy loss is insufficient is the so-called "autoregressive trap" (Zhang et al., 2024). Even small errors in a next-token prediction task can accumulate over long sequences, moving away from the training distribution. Out-of-distribution forecasts then become increasingly worse until the generating distribution completely derails or collapses. This emphasizes the need to evaluate statistics over entire sequences, rather than focusing solely on cross-entropy. A benchmark framework should therefore also measure how fast such errors accumulate by evaluating distributions conditional on the forecasting horizon.

Evaluating generative models in any domain is fundamentally a matter of comparing distributions. Our benchmark performs exactly this task. It reduces a high-dimensional distribution of sequences of order book states $\mathbf{b} \in \mathcal{B}$ and message events $\mathbf{m} \in \mathcal{M}$ to scalars by using scoring functions $\Phi_i : (\mathcal{M} \times \mathcal{B}) \mapsto \mathbb{R}$, $i \in \mathbb{N}$. One-dimensional score distributions can then be compared between real and model-generated data using various norms or divergences $\mathbb{D}$. By estimating the difference between the unconditional real data distribution $p\{\Phi(d)\}$ and the data distribution under the model $\hat{p}\{\Phi(d)\}$, i.e. $\mathbb{D}\left[p\{\Phi(d)\} \,\|\, \hat{p}\{\Phi(d)\}\right]$, different generative models can be ranked on their ability to match features of the data.

To evaluate the magnitude of the "autoregressive trap" the benchmark evaluates error divergence of distributions, conditional on the interval of the forecasting step $t \in \mathbb{N}$, for interval limits $a, b \in \mathbb{N}$: $\mathbb{D}\left[p\{\Phi(d_{t\in[a,b)})\} \,\|\, \hat{p}\{\Phi(d_{t\in[a,b)})\}\right]$. This allows quantifying distribution shift during inference.

Our framework uses both the $L_1$ norm and the Wasserstein-1 distance as loss metrics. To estimate the $L_1$ norm, we first bin the data. As a robust binning algorithm, we use the Freedman-Diaconis rule (Freedman & Diaconis, 1981), which computes the bin width as $2\frac{IQR}{\sqrt[3]{n}}$, where $n$ is the combined sample size and IQR the inter-quartile range of the real and generated data. The $[0, 1]$-scaled $L_1$ norm, also called the *total variation distance*, can then be estimated as:

$$\frac{1}{2}\|p - \hat{p}\|_1 = \sum_{b \in bins} \frac{1}{2} \left| p\left(\frac{b_{count}}{b_{width}}\right) - \hat{p}\left(\frac{b_{count}}{b_{width}}\right) \right|. \tag{1}$$

While the $L_1$ measure has the benefit of being bounded in the interval $[0, 1]$, the Wasserstein-1 distance, or earth mover's distance, as proposed by Rubner et al. (2000), has the advantage of being sensitive to the distance between the scores. To make losses between different scoring functions comparable, we mean-variance normalize the data before calculating the Wasserstein-1 distance.

For equal sample sizes we can compute the Wasserstein-1 distance as follows. Let $\Phi(d_{real})_{(i)}$ be the i-th order statistic of a score computed from a real data sample drawn from $p$ and $\Phi(d_{gen})_{(i)}$ the i-th order statistic using generated data drawn from $\hat{p}$. Then we have:

$$W_1(p, \hat{p}) = \sum_{i=1}^{n} \left\|\Phi(d_{real})_{(i)} - \Phi(d_{gen})_{(i)}\right\|_1. \tag{2}$$

To evaluate a generative model's ability to adapt to different contexts, we also estimate differences between conditional score distributions

$$\mathbb{D}\left[p\{\Phi_1(d) \mid \Phi_2(d)\} \,\|\, \hat{p}\{\Phi_1(d) \mid \Phi_2(d)\}\right]. \tag{3}$$

In this case, $\Phi_2(d)$ is binned into 10 data deciles $b_j$ of the pooled real and generated data. Distance estimates of these

10 conditional distributions are then weighted according to the mean of the estimated density of both distributions. Letting $X = \Phi_1(d)$ and $Y = \Phi_2(d)$, a conditional metric can be evaluated as

$$\sum_{b_j} \frac{\mathbb{D}\left[p(X \mid Y \in b_j) \,\|\, \hat{p}(X \mid Y \in b_j)\right]}{\times \frac{p(Y \in b_j) + \hat{p}(Y \in b_j)}{2}} \qquad (4)$$

This approach addresses a specific type of distribution shift: the variation of scores, $\Phi_1$, across the distribution of another score, $\Phi_2$. For instance, if the conditioning function $\Phi_2$ represents the mean time of messages within a data sequence, this framework allows us to analyze how distribution shifts affect any score of interest, $\Phi_1$, and to assess the generative model's ability to replicate this dynamic behavior accurately.

Our methodology formalizes and naturally extends common evaluation practices for synthetic one-dimensional time series, such as financial returns, which typically emphasize distributional similarity. Our framework enables a quantitative assessment of distributional properties in *structured high-dimensional* time series. By adapting the scoring functions, our approach could also be applied to financial transactions, payment data, streamed price quotes in forex markets, multi-asset limit order books, or decentralized crypto market protocols.

### 4.1. Impact Response Functions

A primary difficulty with historical LOB data is that counterfactual scenarios are impossible to evaluate, as the data do not respond to additional injected orders. Generative models are a unique opportunity to generate a response to counterfactual scenarios as they address this limitation.

It is therefore crucial that such models be evaluated on their ability to provide a realistic response to different events. As an underlying methodology, the seminal work by Eisler et al. (2012) is used as a basis to compare the impact of different event types. This methodology focuses only on the impact of events, which change the price or quantity of the best bid and ask orders (also known as touch orders), which also constitutes a key limitation of the method.

All events which affect the best prices are classified into one of six order types $\pi \in \Pi$: market orders (MO), limit orders (LO) and cancellations (CA), which are further subdivided into those which affect the mid-price, indicated with subscript 1, and those who do not, with subscript 0: $\Pi = \{MO_0, MO_1, LO_0, LO_1, CA_0, CA_1\}$.

Using the convention in *LOBSTER* data, we define the direction $(dir)$ as 1 for events on the bid side and $-1$ on the ask side. The events are given an $\epsilon$ value based on the expected direction of impact on the mid-price they will provoke. In this context, executions are considered market orders.

$$\epsilon = \begin{cases} dir & \text{if event type is MO or LO;} \\ -dir & \text{if event type is CA.} \end{cases} \qquad (5)$$

This allows calculation of the response function – equation (6). This is calculated empirically using the time average $(\langle \ . \ \rangle_T)$ of the change in the sign-adjusted mid-price $p_t = \frac{a_t + b_t}{2}$ following a given event, for different lag-times $l$. The event lag times are chosen to be distributed uniformly on a logarithmic scale between 1 and 200 ticks. The prices are normalized by tick size to enable a comparison between various stocks.

$$R_\pi(l) = \langle (p_{t+l} - p_t)\epsilon_t | \pi_t = \pi \rangle_T \qquad (6)$$

Eisler et al. (2012) identify averaged response functions for 14 random stocks over a period of 53 trading days. Whilst such analysis gives a good baseline to which we can compare our results, for model evaluation we instead directly compare the functions between model-generated and real sequences (matched based on the seeded starting point or the time of day) for individual stocks. Once the response functions are calculated, we create a measure of comparison to obtain a score of dissimilarity:

$$\Delta R_\pi = \frac{1}{L} \sum_{l=1}^{L} |R_\pi^{real}(l) - R_\pi^{gen}(l)|, \qquad (7)$$

which is aggregated across all event types by taking the mean $\Delta R = \frac{1}{|\Pi|} \sum_{\pi \in \Pi} \Delta R_\pi$.

### 4.2. Adversarial Measurement

The concept of adversarial measurement involved developing a pre-trained discriminator capable of effectively distinguishing between true and generated trajectories. This discriminator is a binary classifier, generating a probability estimate of a trajectory being real. We only use the order-book states as input, ignoring the message sequences. The discriminator is trained using two batches of data, each of dimension $(S \times T \times D)$. $S$ denotes the number of sequence samples within the batch, $T$ the length of the orderbook sequences, and $D$ is the dimension of the orderbook state representation.

The discriminator aims to find the "worst-case" function $\Phi^*$ that maximally separates the real and generated distributions by choosing $\Phi^*$ such that it maximizes the divergence between them, i.e., $\Phi^* = \arg\max_\Phi D[p(\Phi(d)), \hat{p}(\Phi(d))]$. This $\Phi^*$, which can be interpreted as a dimensionality reduction operation on the sequence of order book states $\mathbf{b} \in \mathcal{B}$ to a scalar $s$, $\Phi^* : (\mathcal{M} \times \mathcal{B}) \mapsto \mathbb{R}$, can be considered to be an adversarial scoring function. The discriminator attempts to identify the most glaring flaws and differences between

the real and generated samples, distilling these into a single dimension.

Given the sparsity of changes between successive orderbook states, we devised an encoding scheme to optimize the discriminator's performance. An orderbook state comprises the price and quantity from the top $n$ price levels on both the bid and ask sides. In our experiments, $n = 10$ resulting in dimension $D = 40$. Changes in the orderbook state are typically triggered by events that affect a single price-quantity pair. To achieve a more concise, yet informative, representation of the discriminator network, we chose to represent the orderbook based on these changes. Thus the book states $\mathbf{b} \in \mathcal{B}$ and message events $\mathbf{m} \in \mathcal{M}$ map to three-dimensional vectors through $i \in \mathbb{N}$ functions $\Psi_i : (\mathcal{M} \times \mathcal{B}) \mapsto \mathbb{R}^3$. These changes encompass each change in the mid-price, the relative price level where the change occurs, and the corresponding change in quantity. Our discriminator utilizes a 1D convolutional neural network (Conv1D) (Lecun & Bengio, 1995; Kiranyaz et al., 2019) as a feature extractor, followed by an attention mechanism (Vaswani et al., 2017) to capture long-term dependencies across the time steps. Empirical results show that this model, trained and tested on GOOG data from 2023, achieves a Receiver Operating Characteristic (ROC) score of 0.83, indicating that the generated data can be discriminated reasonably accurately. However, the baseline model's performance for GOOG and INTC was poor, with a discriminator ROC score of around 1, indicating significant room for future model improvement. High discriminability may result from model errors, as indicated by imperfect model scores (see Section 6 and Figures 13ff). A distributional mismatch in a single scoring function can be sufficient to make fake data identifiable. To mitigate this issue, future research could evaluate adversarial performance by training a discriminator on perturbed data and reporting scores conditioned on the noise level, particularly as models improve on this benchmark.

### 4.3. Mid-price Prediction

To evaluate the potential value added from additional generated synthetic data, we measure the impact of the additional training data on prediction quality for a simple downstream task. Concretely, we adapt the LOBCAST (Prata et al., 2024) implementation to train an MLP with one million parameters to classify the mid-price movement for different prediction horizons. The MLP is trained to predict three classes: *up*, *down*, and *stationary*. Following Prata et al. (2024), classes are defined based on the movement of the mid-price over a threshold value. For each generative model evaluated, two separate MLPs are trained, one on only real data, and another on real & generated data. F1-scores calculated on held-out real data are then evaluated as a measure of the impact of using the generated data in training.

## 5. LOB-Bench Package

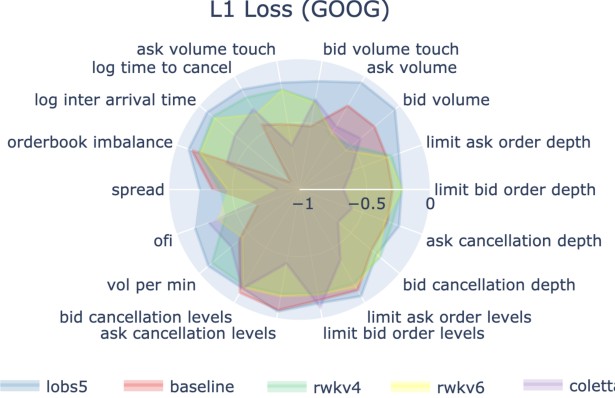

Figure 3: **Model comparison spider plot:** the *LOBS5* model beats the *baseline* and *coletta* model on almost all scores. Note: the radial axis is inverted by plotting the negative loss (larger is better).

Based on the evaluation framework outlined in section 4, we develop a Python benchmark package, allowing for a convenient and comprehensive evaluation of generated LOB data. The benchmark is highly customizable, as scoring functions $\Phi$ can easily be added, removed, or modified, and provides a standardized model comparison using the default scoring functions provided. The benchmark reports aggregate model scores by computing the mean, median, and inter-quartile mean (IQM[1]) across all conditional and unconditional scoring functions, along with bootstrapped confidence intervals.

The benchmark performs both unconditional and conditional evaluation of generated data, by computing distributions of statistics of interest conditionally on the value of another statistic. To evaluate the magnitude of the effect of error divergence or "snowballing errors," distributions are also evaluated conditional on the prediction horizon. Distributional accuracy is measured by computing the $L_1$-norm and Wasserstein-1 distance between the real and generated distributions. Specific supported examples of more complex conditional distributions are the response functions, describing the distribution of events conditional on other events having occurred at a certain prior lag. As these distributions usually have high variance, and to be consistent with the extant literature, we instead measure mean absolute differences in their means for a range of lags to evaluate market impact curves.

We include multiple conditional scoring functions from the finance literature, for example, ask volume conditional on the spread, the spread conditional on the hour of the day, and the spread conditional on the volatility of 10ms returns.

---

[1]mean of all values between the 25. and 75. percentile

| Statistic | Description |
|---|---|
| Bid-Ask Spread | Difference between the highest price a buyer is willing to pay (the bid) and the lowest price a seller is willing to accept (the ask) |
| Order Book Imbalance | Imbalance for the best prices is computed as $(\text{bid size} - \text{ask size}) / (\text{bid size} + \text{ask size})$ |
| Message Inter-Arrival Time | Time between successive order book events (on a log-scale due to a long right tail) |
| Time-to-Cancel | Time between submission and first (partial) cancellation for cancelled limit orders, measured on a log-scale. |
| Bid/Ask Volume | The volume of all orders on the bid, respectively ask, side of the LOB. We also evaluate the volume only at the best price levels. |
| Limit & Cancellation Depths | Absolute distance of new limit orders or cancellations from the mid-price |
| Limit & Cancellation Levels | The price levels at which events occur $\in \mathbb{N}$ |
| Volume per Minute | Traded volume in one-second intervals, scaled to a minute. |
| Order Flow Imbalance (OFI) | Metric from (Cont et al., 2012) considering the imbalance in submitted orders for a rolling window of messages. |
| OFI (Up/Stay/Down) | OFI (see above), conditional on the subsequent message's mid-price move: Up/Static/Down |

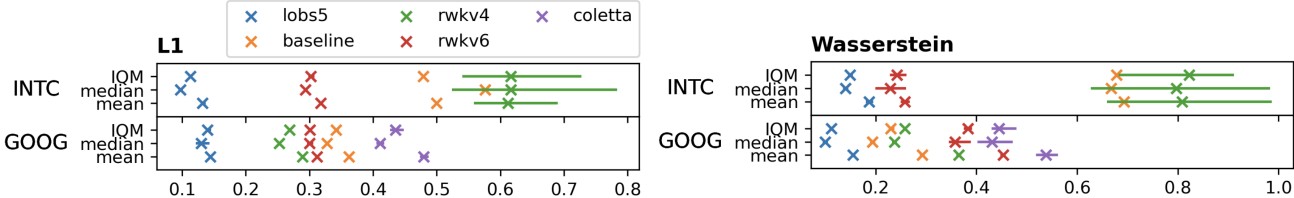

Figure 4: **Model score summaries** (lower is better). The *LOBS5* model achieves the lowest overall scores. *Coletta* beats the *baseline* on the Wasserstein metric, but not for $L_1$. Error bars are bootstrapped 99% CIs.

## 6. Results

As a first test case for our benchmark, we adapt the autoregressive state-space model using S5 layers (Gu et al., 2021) from Nagy et al. (2023) (*LOBS5*). Particularly, we scale up the model size to 35 million parameters and more than double the training period to the entire year of 2022.

We also evaluate data generated by the models from Cont et al. (2010) (*baseline*), Coletta et al. (2022) (*Coletta*) and models based on Peng et al. (2023; 2024) (*RWKV 4* and *6*). The baseline model, which employs parametric arrival processes, is adapted to generalize across both small and large tick limit order book (LOB) dynamics by utilizing estimated empirical arrival rates directly, rather than fitting a power law. Additionally, we infer data features present in *LOBSTER*, such as individual message IDs, which are not generated by Cont et al. (2010). This inference is particularly important for capturing order cancellations, as we uniformly sample target limit orders from the available orders at the specified price level. For the *Coletta* model, we implement a

*LOBSTER* data interface to facilitate the conversion of data formats. For the *RWKV* models, we apply autoregressive next-token prediction, but on a larger model (170 million parameters), without any data pre-processing and using an off-the-shelf byte-pair tokenizer (Sennrich et al., 2016), as is used for LLMs. These models are trained solely on message data, without *any* order books, which do not require propagation of a calculated orderbook state, unlike in Nagy et al. (2023). The S5, RWKV, and baseline results presented here are computed, following Nagy et al. (2023), for Alphabet (GOOG) and Intel (INTC) stock on a sub-sample of the test data from January 2023. The *Coletta* model is trained on three days from January 2019 and tested on three subsequent days, following the procedure in Coletta et al. (2022), which is necessary due to the high computational cost for training and inference with *Coletta*. Comparing all models, we observe that the *LOBS5* model provides state-of-the-art performance on the benchmark task.

Figure 3 presents a key benchmark feature to compare multiple models across multiple score dimensions, allowing an examination of individual strengths and weaknesses. To provide summary scores per model, Figure 4 reports the mean, median, and inter-quartile mean for the L1 and Wasserstein-1 metrics for all models [2]. Error bars demarcate the 99%

[2]The *Coletta* model (Coletta et al., 2022) was trained on both

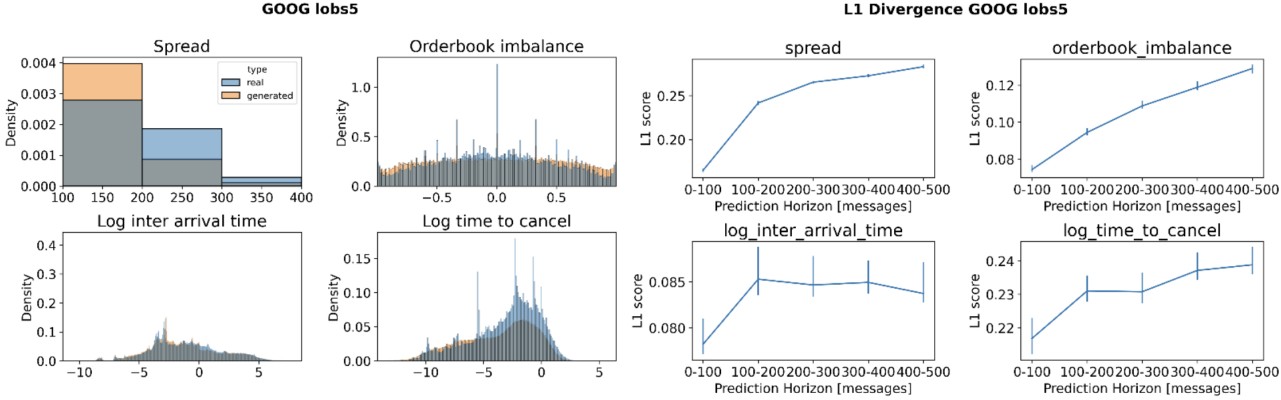

Figure 5: *LOBS5* **results** – (left): histogram matching of unconditional score distributions for real and generated data. (right): error accumulation – the further out the prediction horizon, the worse is the model performance – an important model characteristic to measure.

bootstrapped confidence intervals. Metrics for individual scoring functions are shown in Figure 12 (Appendix E).

The benchmark also measures error divergence by comparing distributions of scoring functions, conditional on the inference time step. These demonstrate the rate at which distributions diverge from real data. Results show increasing errors across all models with the fastest divergence exhibited by the RWKV models across most scores. For the S5 model in particular, scoring functions with a dependence on features of the book states, which only gradually change, such as book volume, are expected to diverge, as the initial real data seed decays. However, the rate of decay can still be compared between models. See Figure 18 in appendix E for L1 divergence curves and appendix D for a discussion of and an ablation experiment on the effect of bin sizes on divergence scores.

The response functions for Alphabet (GOOG) are shown in Figure 7 for the *Baseline* and *LOBS5* models. The *LOBS5* model generally reproduces curves similar to real data but does so better for small-tick stock GOOG. In contrast, the *baseline* model (Cont et al., 2010) cannot faithfully reproduce impact curves. We do not post the impact curves for the *Coletta* and *RWKV* models, as they quickly diverge due to error accumulation in inference. The average $L_1$ distance between real and generated impact curves (see equation (7)) is $\Delta R = \mathbf{2.45}$ for *LOBS5*, and $\Delta R = \mathbf{126}$ for *RWKV-6*. For the *LOBS5* model, we observe differences mainly in the $MO$ orders at short lags. This is due to the JAX-LOB simulator (Frey et al., 2023), which is used by the *LOBS5* model during inference. JAX-LOB splits limit orders, which can only be partially filled, into

GOOG and INTC data, but failed to produce reasonable results for INTC, which is expected since the model was designed for small-tick stocks, whereas INTC is not.

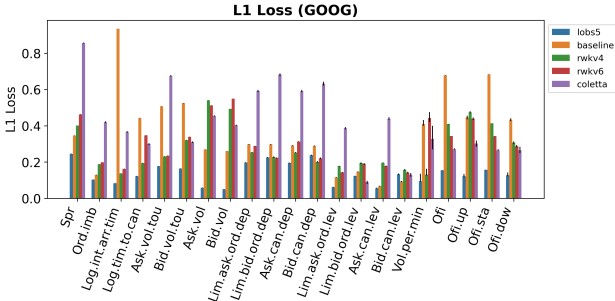

Figure 6: $\mathbf{L_1}$ **distance** between real and generated data histograms (incl. 99% CIs). *baseline* performs well on LOB depth and level-related scores, and much worse on time and volume metrics. *LOBS5* dominates L1 loss for GOOG.

execution messages and additional resting limit orders, thereby merging a multilevel midprice change into a single order book update.

F1-scores for mid-price trend forecasting using a simple MLP, following the LOBCAST implementation (Prata et al., 2024), are illustrated for Alphabet (GOOG) in Figure 8. We observe that, for the Coletta model in particular, including generated synthetic data markedly reduces the resulting prediction precision and recall. This is in line with model rankings based on the distributional distances shown in Figure 5. Generally, the same pattern is also present across the other models, particularly for short prediction horizons, although to a less extreme extent. Mixing generated data from either the baseline or LOB-S5 models into the training set has no significant effect for longer prediction horizons. Although it is desirable for models to generate data which increase prediction performance, not acting detrimentally

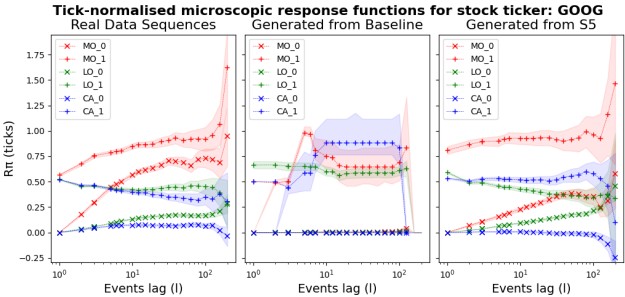

Figure 7: Comparison of tick-normalized **mid-price impact response functions** for different event types between real and generated data-sets. Shaded regions are 99% confidence intervals. Compared are the *LOBS5* and stochastic *baseline* models. In contrast to the *baseline*, *LOBS5* reproduces most features of the expected impact response functions.

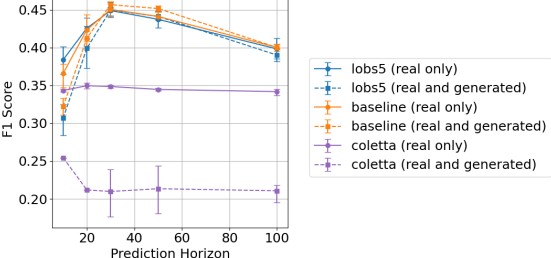

Figure 8: Comparison of **F1-scores for the MLP classification models** trained on historical (real) data only (solid lines), and trained on both real data and data generated (dashed lines) for a subset of the generated models. The error bars depict the 95% confidence intervals bootstrapped from five different training seeds.

is a minimum expectation. The results show that current generative models are not yet able to generate data aiding this mid-price trend prediction task. Beyond this conclusion, further quality differentiation between the models based on this task is limited. In contrast, the distributional LOB-Bench evaluation results provide a clearer picture of model strengths and potential for future improvement.

## 7. Conclusions

We introduce LOB-Bench, an evaluation framework for generative AI models for order-book modelling. Crucially, our framework contains analysis tools that make it easy for users across the machine learning and finance domains to benchmark their message-level order-flow models.

We believe that LOB-Bench will greatly facilitate core ML research working on sequence modelling to apply their innovations to this challenging and relevant real-world problem

while also making it easier for finance practitioners to use best-practice tools.

One of the interesting aspects of generative AI models for microstructure data is the ability to model counterfactuals, which is closely related to the notion of price impact in financial modelling. Factoring in the reactions of other market participants to one's actions with conventional approaches is very challenging, but our benchmark suite for generative LOB models provides extensive tests to evaluate whether generated data reproduces the expected response functions at a larger scale. Future research could involve measuring the extent to which generative models match the market impact laws in the literature, such as the "square root law" (SRL) (Tóth et al., 2016). We hope that LOB-Bench opens the door to many new studies, including the training of reinforcement learning algorithms and multi-agent models for tasks such as trade execution, with the ability to model realistic reactions of different market participants.

## Acknowledgements

We gratefully acknowledge the *Oxford-Man Institute of Quantitative Finance* and the *Foerster Lab for AI Research* for providing us access to their GPU compute servers. These computational resources were instrumental in carrying out the experiments and analyses presented in this work. AC acknowledges funding from a UKRI AI World Leading Researcher Fellowship (grant EP/W002949/1. AC and JF acknowledge funding from a JPMC Faculty Research Award. For the purpose of Open Access, the authors have applied a CC BY public copyright licence to any Author Accepted Manuscript version arising from this submission.

## Impact Statement

This paper presents work whose goal is to advance the field of machine learning. There are many potential societal consequences of our work, none which we feel must be specifically highlighted here.

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

## A. Benchmark Code

The benchmark code can be found on GitHub at https://github.com/peernagy/lob_bench, additional resources can be found on the project website https://lobbench.github.io/.

The benchmark suite provides a convenient API functionality to evaluate model data for a range of scoring functions and metrics. A specification of such functions and loss metrics can be defined in a configuration dictionary, which can then be passed to a function performing the unconditional and conditional model evaluation. Similarly, the benchmark provides functions to compute the market impact curves, along with a mean L1 score. A default configuration dictionary, specifying the scoring functions reported here, evaluated using L1 and Wasserstein-1 loss, is similarly provided for easy reproducibility.

To run the benchmark, real and generated data sequences must be stored in LOBSTER format [3] as csv files. Files must be separated by real data, generated data, and (real) data used to condition the generation. A more detailed description can be found on GitHub.

## B. LOBS5 Training Details

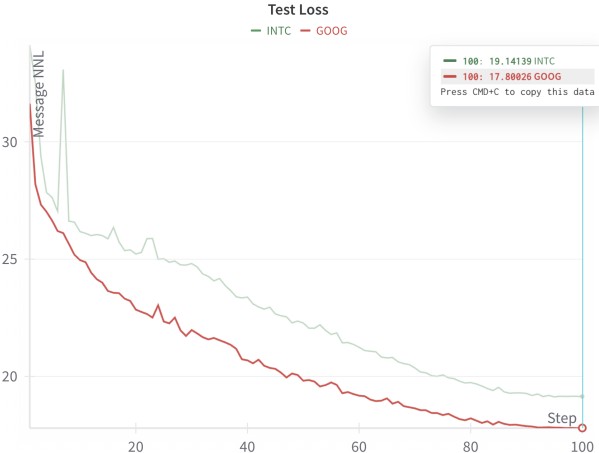

Figure 9: Test set (2023 data) loss curves for the LOBS5 model, measuring the mean *per-message* negative log-likelihood for INTC (green) and GOOG (red) throughout 100 training epochs. Message cross-entropy after 100 epochs is 19.14 for INTC and 17.80 for GOOG.

Starting from the model introduced by Nagy et al. (2023), we have scaled up the model size by adding additional S5 layers, with a resulting parameter count of approximately 35M (compared to originally 6.3M). The training consisted of 100 epochs of shuffled data sequences from the entire year of 2022, training with a total training budget of 30.4 L40 days (3.8 days across 8 GPUs). Adam (Kingma & Ba, 2014) was used as an optimizer with a cosine learning rate schedule. Losses on the test set over the course of training are displayed in Figure 9.

With this larger model, we also successfully removed the explicit error correction mechanism, which originally rejected semantically incorrectly generated messages, as error rates could be sufficiently reduced by scaling the model.

---

[3]https://lobsterdata.com/info/DataStructure.php

## C. RWKV Training Details

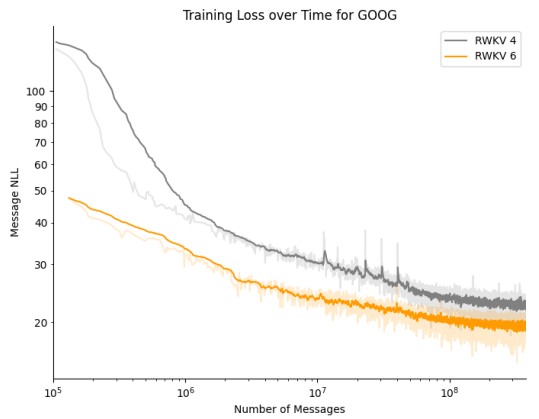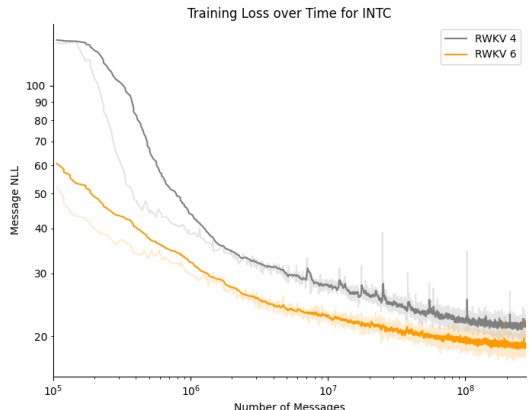

Figure 10: Training loss curves for RWKV model training. The y-axis represents the average negative log likelihood of the messages being trained on, calculated as the sum of negative log likelihoods of each token in the message. The bold lines represent the exponential moving average of the true curve (presented with lower opacity) with an $\alpha$ of 0.1 to better highlight the trend.

We trained all RWKV models using an autoregressive training scheme with only message data (i.e., without any orderbook state information), and apply the same message filtering protocol as Nagy et al. (2023). We first tokenized each LOBSTER dataset using a byte pair encoder (Sennrich et al., 2016) trained on GOOG 2017 messages, resulting in datasets of 5.5 billion tokens for INTC 2022 (corresponding to 276 million messages) and 7.5 billion tokens for GOOG 2022 (corresponding to 380 million messages). We divide each dataset into chunks of 16384 tokens, and randomly shuffle these chunks for training.

For training, we initialize the parameters from the open source base pretrained RWKV models, each consisting of 170 million parameters, and train them on 8 chunks per optimization step, using the DAdapt-AdamW optimizer (Defazio & Mishchenko, 2023) in Optax (DeepMind et al., 2020), without scaling the learning rate or using any learning rate schedulers. For stability, we clipped the maximum global gradient norm to 1.0 (Pascanu et al., 2013). In total, training all 4 of our RWKV models (2 model architectures, each over 2 datasets) took 10 L40S days. We present loss curves in Figure 10.

## D. Sensitivity of Divergence Metrics to the Bin Size

We adopt a dynamic bin size determined by the Freedman–Diaconis (FD) rule, which is specifically designed to adapt to the underlying data distribution. As a result, we do not anticipate significant sensitivity to the choice of bin size. This choice is further supported by a theoretical convergence property: the FD rule minimizes the integrated mean squared error (IMSE) between the histogram and the true data distribution (Freedman & Diaconis, 1981).

To empirically evaluate the impact of different bin sizes on the reported divergence scores, we conduct the following experiments. The same divergence metric computations are repeated with half the regular bin size, as well as with double the regular bin size. To illustrate low bin size sensitivity, we perform this experiment for the LOBS5 model across all evaluated divergence metrics. We note that smaller bin sizes tend to increase both L1 and Wasserstein-1 errors, as data points, which would have been grouped into a single bin, could then be part of two separate bins. On the contrary, larger bin sizes tend to decrease errors, as data points close to each other are more likely to be grouped together.

To empirically assess the robustness of our results to bin size variation, we repeat the divergence metric computations using both half and double the default FD bin size. This analysis is conducted using the LOBS5 model across all evaluated divergence scores. We observe that reducing the bin size generally increases the $L_1$ errors. This is likely because data points that would have previously fallen within the same bin may now fall into separate bins, amplifying divergence estimates. Conversely, increasing the bin size tends to reduce these errors, as nearby data points are more likely to be aggregated into the same bin, leading to smoother approximations.

Figure 11 shows the mean divergence scores for the LOBS5 model, along with 99% confidence intervals. Doubling or

halving the bin size results in a consistent decrease or increase in error scores, respectively. Notably, the magnitude of these deviations is comparable to the width of the confidence intervals under the default FD bin size. Given that these changes are relatively small and systematic, we conclude that while the choice of a theoretically grounded binning rule is important, model ranking remains stable as long as the same binning strategy is applied uniformly across all evaluated models.

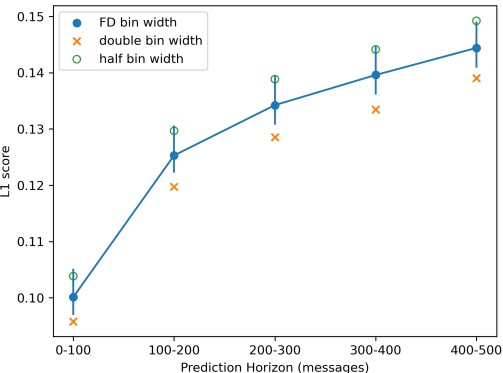

Figure 11: L1 Divergence Scores with half and double regular bin size. Large bin size deviations (halfing or doubling) systematically affect the level of scores but not their ranking.

## E. Additional Figures

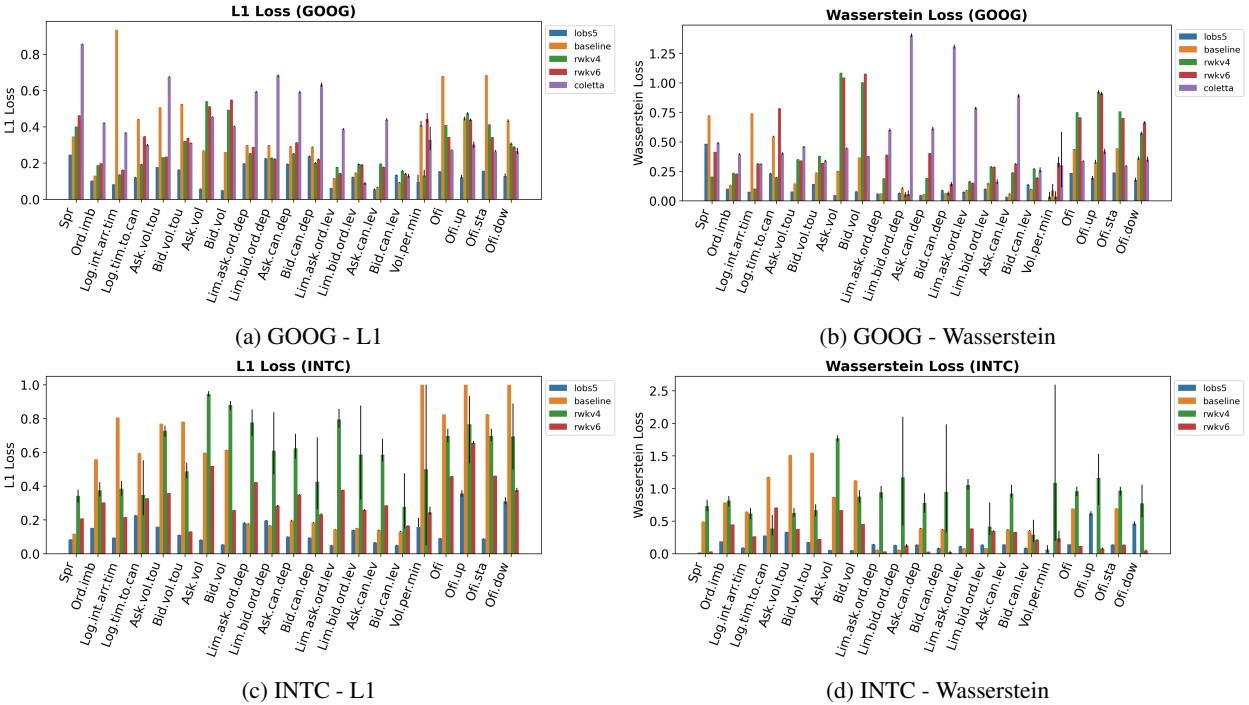

(a) GOOG - L1

(b) GOOG - Wasserstein

(c) INTC - L1

(d) INTC - Wasserstein

Figure 12: L1 and Wasserstein-1 errors of generated unconditional distributions for easy comparison between Alphabet (GOOG) and Intel (INTC). Error bars show 99% bootstrapped confidence intervals.

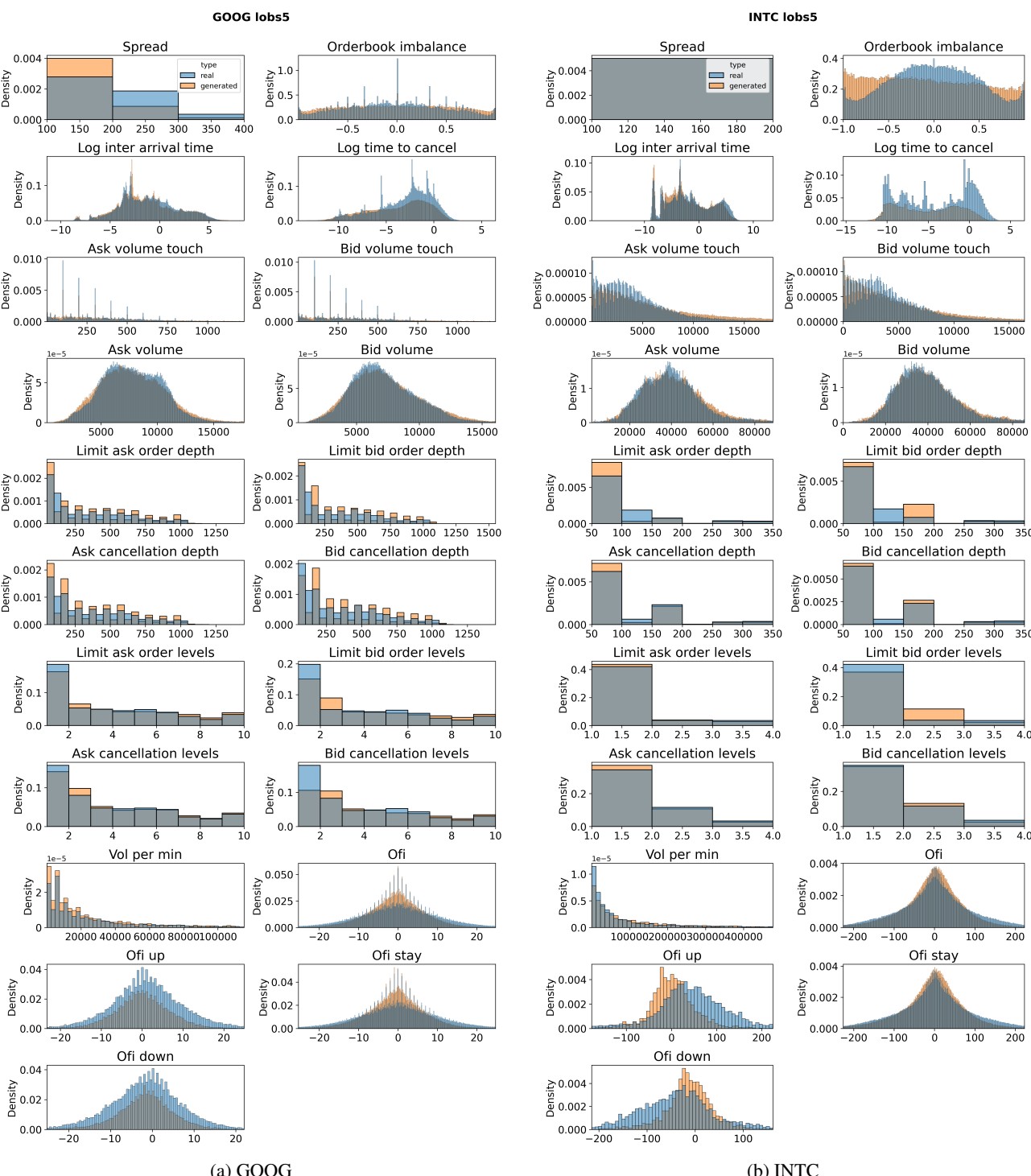

Figure 13: *LOBS5* - histograms comparing score distributions for real (blue) and generated (orange) LOB data for Alphabet (GOOG) and Intel (INTC) stocks. Overall, the generative *LOBS5* model evaluated here, adapted from Nagy et al. (2023), does a good job in matching data along various dimensions. Bigger errors in matching distributions are visible in e.g. spread (GOOG), orderbook imbalance (INTC) and time to cancel (GOOG and INTC).

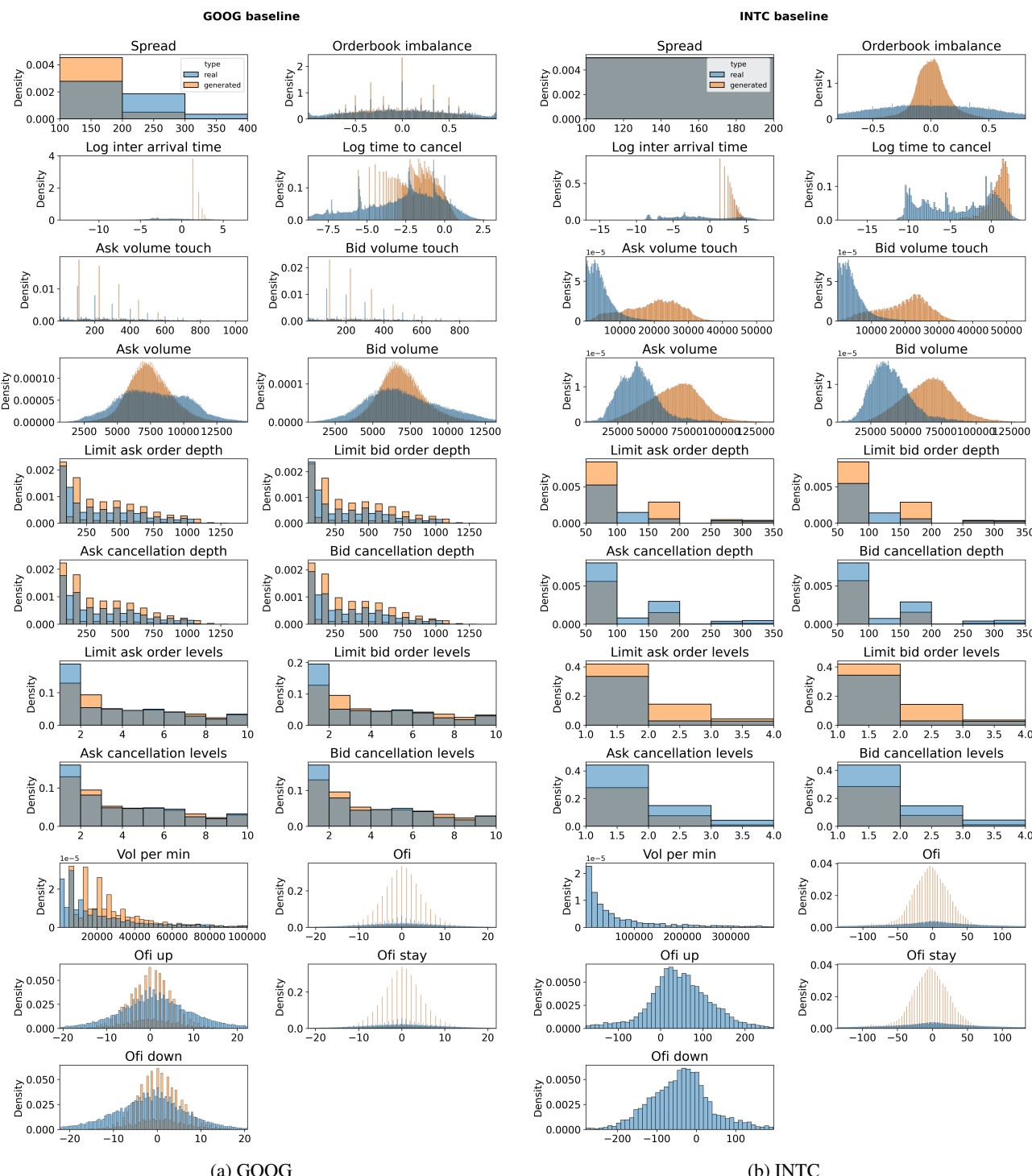

(a) GOOG

(b) INTC

Figure 14: *baseline* - histograms comparing score distributions for real (blue) and generated (orange) LOB data for Alphabet (GOOG) and Intel (INTC) stocks. The (Cont et al., 2010) model does a decent job matching some of the scores, particularly discrete ones, such as depths and levels. Clear shortcomings are visible in scores such as orderbook imbalance or volumes.

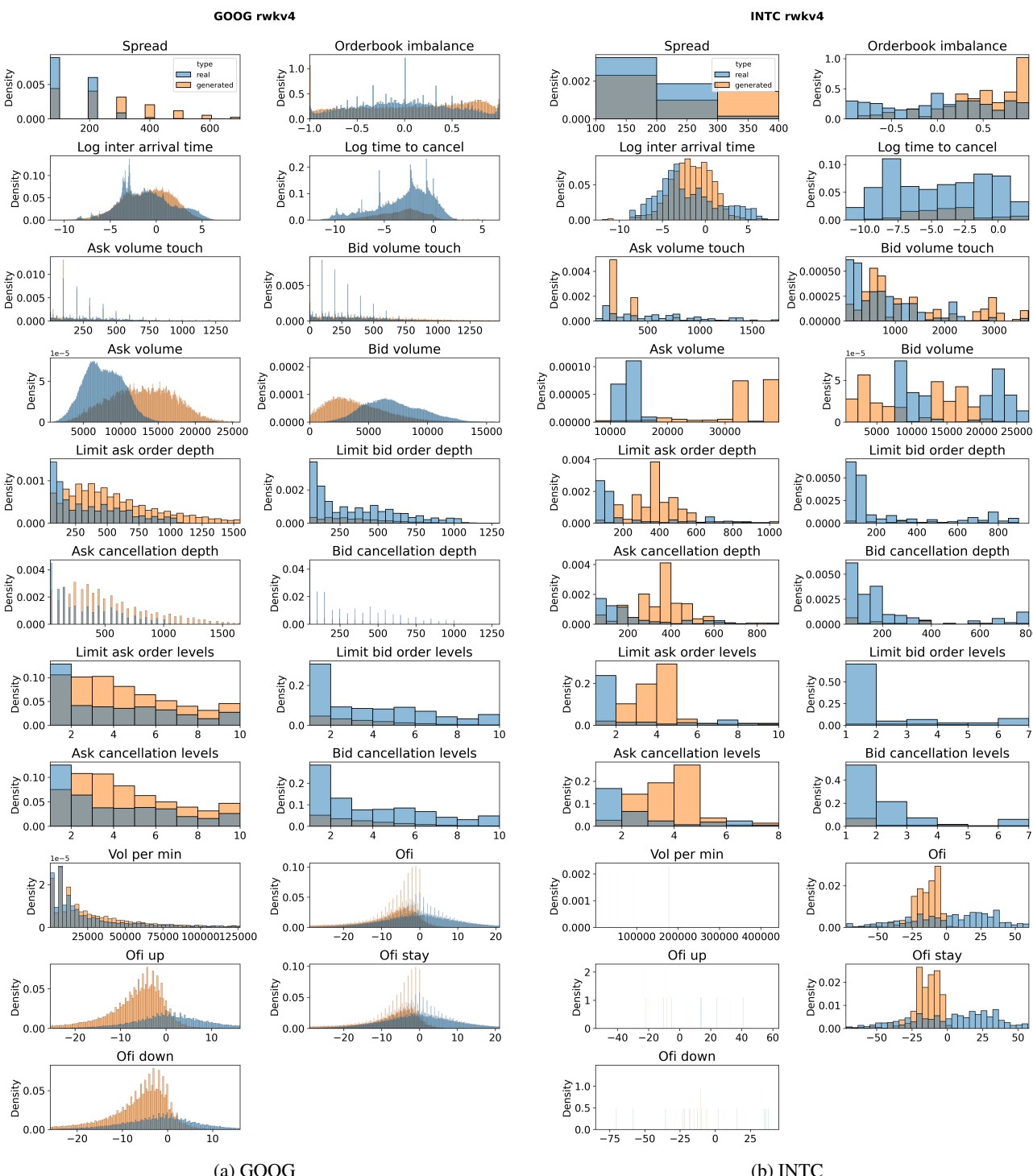

(a) GOOG

(b) INTC

Figure 15: *rwkv4* - histograms comparing score distributions for real (blue) and generated (orange) LOB data for Alphabet (GOOG) and Intel (INTC) stocks. The model produces volatile data with larger spreads, missing correct order levels, leading to difficulty matching book volumes.

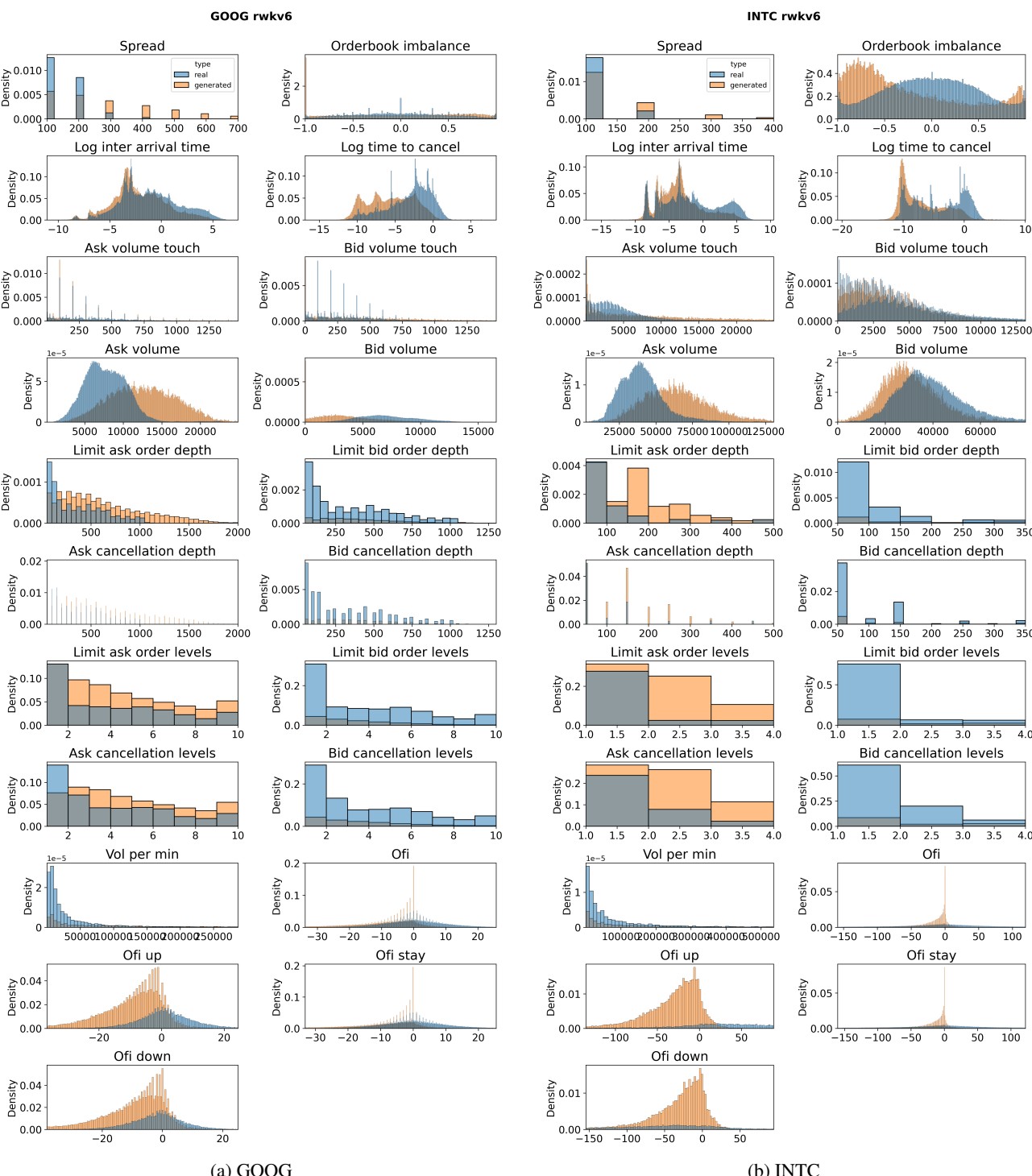

(a) GOOG

(b) INTC

Figure 16: *rwkv6* - histograms comparing score distributions for real (blue) and generated (orange) LOB data for Alphabet (GOOG) and Intel (INTC) stocks. The model has similar shortcomings to RWKV 4 (wrong price levels, mismatched book volumes etc.) due to tokenization of raw data and missing order book information.

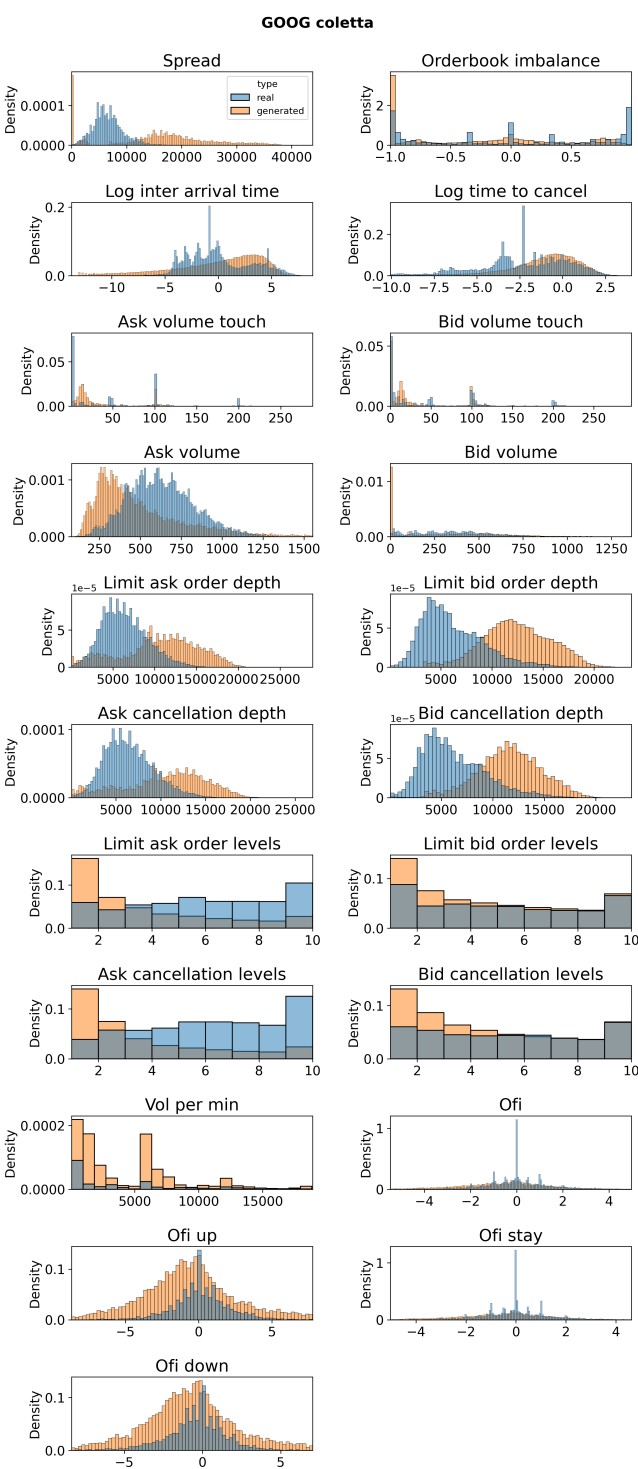

Figure 17: *coletta* - GOOG - histograms comparing score distributions for real (blue) and generated (orange) LOB data for Alphabet (GOOG) and Intel (INTC) stocks.

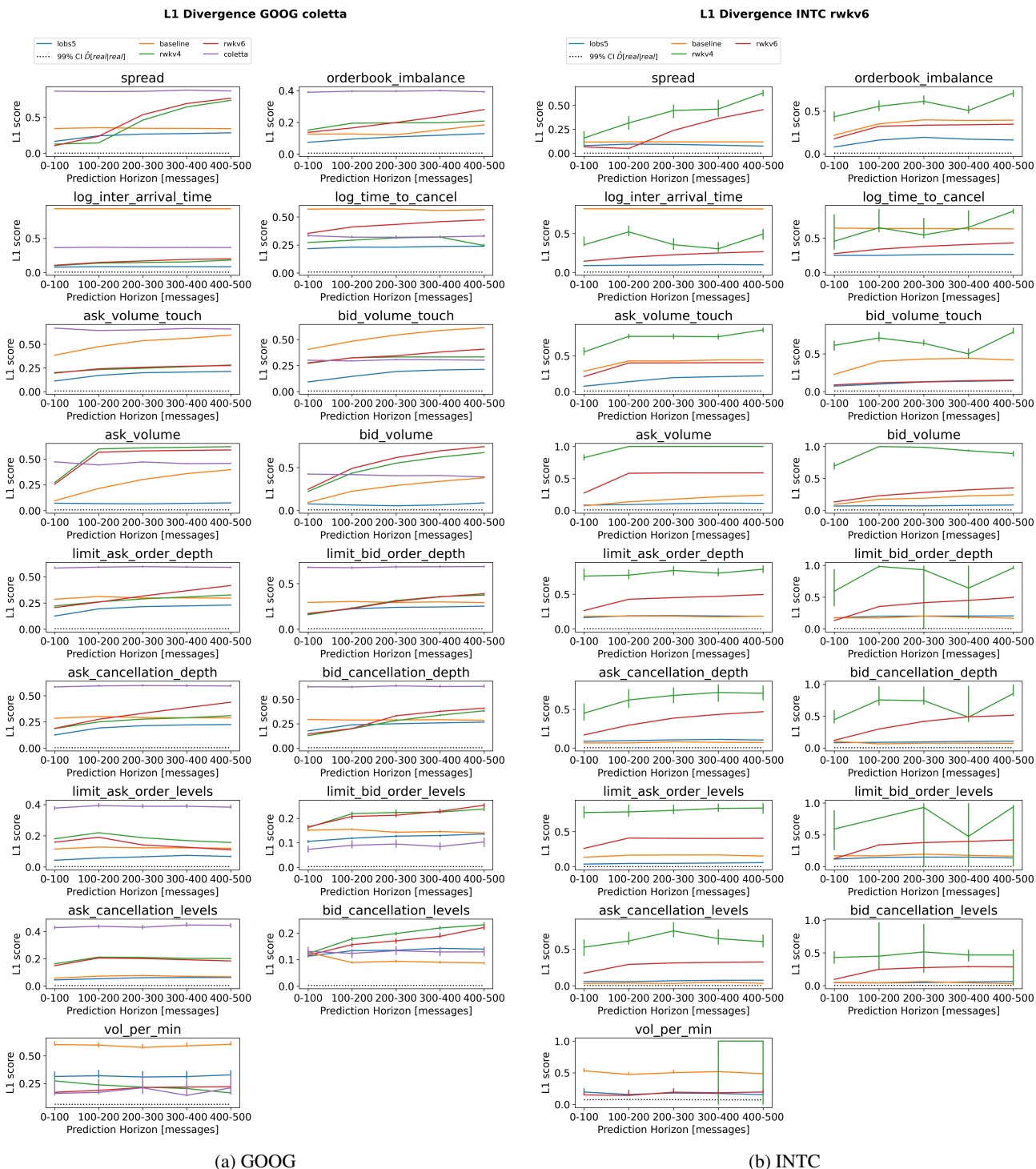

(a) GOOG

(b) INTC

Figure 18: L1 error divergence: comparing the L1 errors of score distributions of real data with generated data distributions at a specific horizon into the future shows accumulating model errors. This is explainable due to snowballing errors caused by teacher forcing (conditional next token loss). A good model should be able to control errors for sequence lengths as long as possible. To provide a significance threshold over pure sampling noise, the dotted lines plot the 99. percentile of L1 error between bootstrapped samples of only real data.

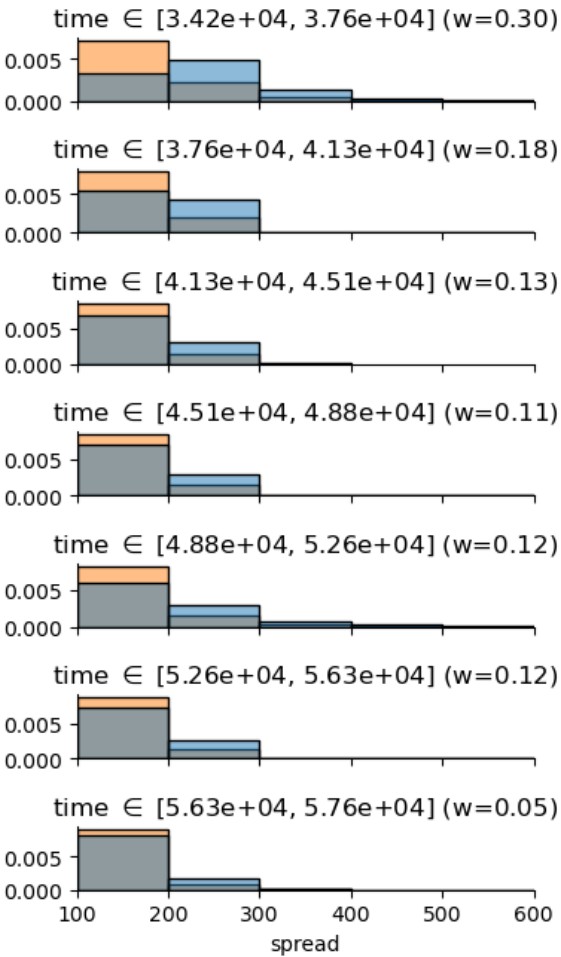

(a) Bid-ask spread conditional on the hour of the day: spreads are higher early in the day, where the generated data also exhibits too narrow spreads.

(b) Spread conditional on volatility: higher volatility corresponds to higher frequency of higher spreads. The model does not fully capture this change, as the higher discrepancy in high-volatility bins shows.

Figure 19: Histograms of conditional score distributions for real (blue) and generated (orange) data for the Alphabet stock (GOOG). Weights $w$, expressing the share of data in the bin, measure the impact of the specific conditional distribution (row) on the total metric loss.

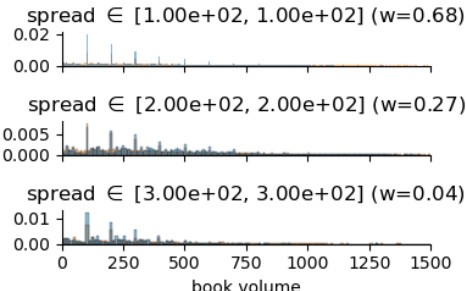

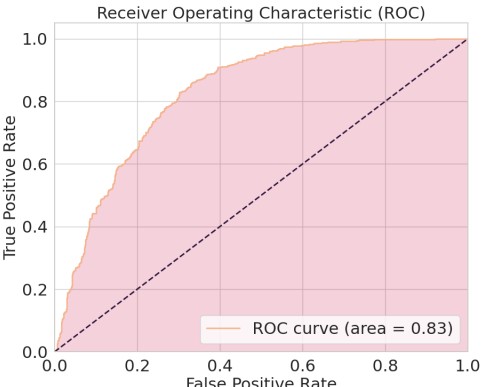

Figure 20: Histograms of total book volume conditional on bid-ask spread for Alphabet stock (GOOG). Weights $w$, expressing the share of data in the bin, measure the impact of the specific conditional distribution (row) on the total metric loss.

Figure 21: LOBS5 - ROC curve of the discriminator on test data (GOOG). The discriminator represents a worst-case adversarial score function by learning to effectively differentiate between real and generated sequences of LOB states.

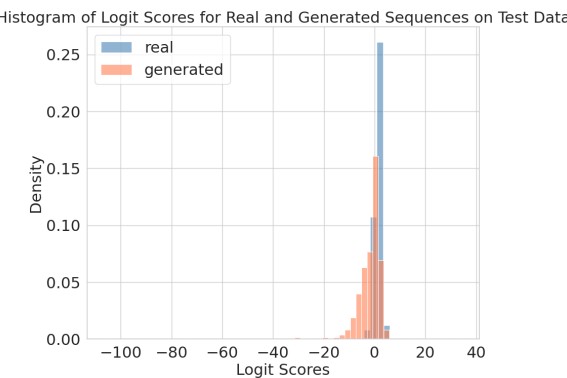

Figure 22: LOBS5 - Histogram of logit scores for real and generated sequences on held-out test data (GOOG). Matching this distribution well would indicate high model quality, as even a trained discriminator network would not be able to differentiate the distributions.

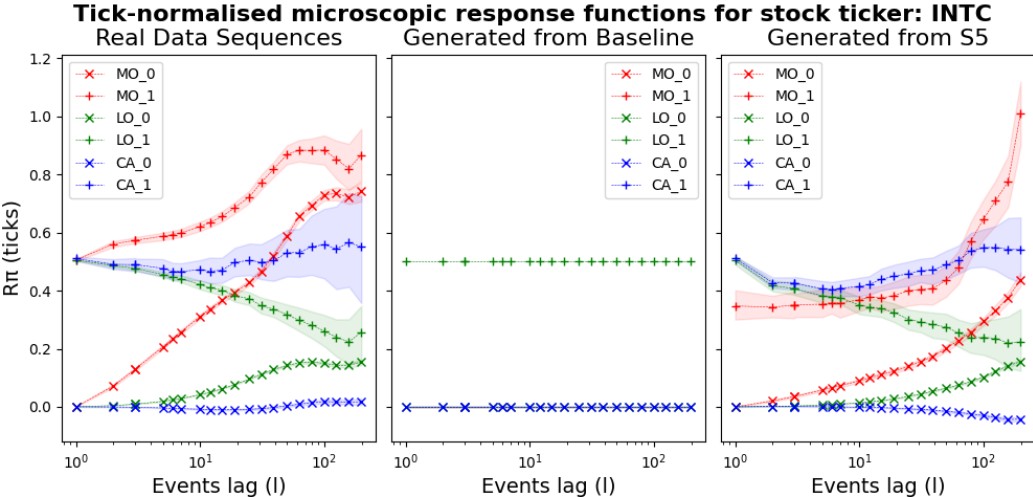

Figure 23: Comparison of impact response functions for different event types between real and generated data-sets, tick-normalized mid-price response. Shaded regions are 99% confidence intervals. There is a comparison between two select models: the LOBS5 and the stochastic baseline. We see that, in contrast to the baseline, the generative model is able to reproduce much more of the expected impact function, though not as well as for GOOG.

