# OpenReview forum: "LOB-Bench: Benchmarking Generative AI for Finance - an Application to Limit Order Book Data"
_ICML.cc/2025/Conference — ICML 2025 poster_

### Official Review · Reviewer_Nn9N · 2025-03-12

**Overall Recommendation:** 3

**Summary:**

The paper introduces LOB-Bench, a benchmark designed to evaluate the quality of generative models for limit order book (LOB) data. The authors propose a quantitative evaluation framework that measures distributional differences between generated and real LOB data. LOB-Bench assesses key LOB metrics such as spread, order book volumes, order imbalance, and market impact using unconditional and conditional statistical comparisons with L1 norm and Wasserstein-1 distance. It also incorporates adversarial evaluation via a discriminator network to distinguish real from synthetic data. The study benchmarks various generative models, finding that the LOBS5 model outperforms traditional approaches. It accurately replicates price impact functions, while classic LOB models fail in this aspect.

**Claims And Evidence:**

The claims made in the submission are largely supported by quantitative evaluation methods, comparative analyses, and empirical results.

**Essential References Not Discussed:**

Not to my knowledge.

**Experimental Designs Or Analyses:**

The results section evaluates generative models for limit order book (LOB) data using the LOB-Bench framework, comparing LOBS5, baseline, Coletta, and RWKV models. As mentioned before, one specific concern is that there is a lack of robustness across different market conditions. The models are tested on only two stocks (GOOG & INTC), limiting insights into their performance across different asset classes, market regimes, and varying liquidity conditions. Expanding the benchmark to include volatile stocks, ETFs, and multi-market scenarios would improve generalizability.

**Methods And Evaluation Criteria:**

The proposed methods and evaluation criteria in LOB-Bench make sense for evaluating generative AI models in the limit order book (LOB) modeling context. More specifically, the unconditional and conditional distributional comparisons provide a comprehensive statistical framework for evaluating how closely generated LOB data resembles real market data. Metrics like L1 norm and Wasserstein-1 distance effectively measure distributional accuracy across different time horizons and order book features.

However, the benchmark is tested on Alphabet (GOOG) and Intel (INTC) stocks, but it is unclear whether the results generalize to other stocks which can be more volatile. Furthermore, the framework focuses on statistical similarity but does not evaluate how well synthetic data supports real-world financial applications like algorithmic trading backtests and market stability simulations.

**Other Comments Or Suggestions:**

Minor improvement on grammar:
1. "... which is explainable as it was intended for small-tick stocks, which INTC is not." -> "... which is expected since the model was designed for small-tick stocks, whereas INTC is not."
2. "passed to function" -> "passed to a function"
3. "The mode produces..." -> "The model produces..."

**Other Strengths And Weaknesses:**

The study focuses primarily on distributional similarity metrics (L1 norm, Wasserstein-1) and market impact curves but does not evaluate how well the generated data supports actual trading strategies. Without backtesting in a simulated or real trading environment, it’s unclear if these generative models can improve decision-making for market participants.

**Questions For Authors:**

The following questions are listed in descending order of importance:

1. How generalizable is the evaluation system across different stocks and market regimes?

2. The paper introduces distributional evaluation metrics (L1 norm, Wasserstein-1 distance) to compare real and generated LOB data. How sensitive are these metrics to different market conditions (e.g., high volatility vs. low volatility periods)? Have you considered alternative evaluation metrics. For instance, evaluating models trained with synthetic data using the real data?

3. The paper suggests that LOB-Bench-generated data could be useful for reinforcement learning (RL) training. Have you tested any RL-based trading strategies using synthetic data, and how well did they generalize to real market conditions?

**Relation To Broader Scientific Literature:**

This paper builds on and extends multiple areas of research in financial market simulation and generative AI. It directly connects to the work of Vyetrenko et al. (2019) on realism metrics for limit order book (LOB) market simulations, which established the importance of evaluating synthetic financial data against empirical LOB properties. While Vyetrenko et al. relied on agent-based models (ABMs) to replicate market behavior, LOB-Bench focuses on deep learning-based generative models, benchmarking S5 state-space models, RWKV transformers, and GANs. This work is also closely related to Coletta et al. (2023), which explored conditional generative adversarial networks (CGANs) for LOB environments, emphasizing market reactivity and stylized fact replication. While CGANs aim to simulate realistic order flows, LOB-Bench goes further by introducing quantitative evaluation metrics (L1 norm, Wasserstein-1 distance) and adversarial discrimination to measure the closeness of synthetic and real data systematically. Ultimately, this work has the potential to establish a foundation for developing more robust and interpretable generative models in high-frequency trading.

**Theoretical Claims:**

The paper does not involve theorems.

---

> ### Author Rebuttal · Authors · 2025-04-01
>
> We sincerely thank reviewer Nn9N for their detailed remarks and for recognizing the potential of this work in establishing a foundation for developing more robust and interpretable generative models in finance. We address the queries and concerns below, and would welcome any follow-up questions or suggestions. If these have been suitably addressed, we would appreciate an increase in the review score to help build a stronger case for acceptance.
>
> ## Generalizability to Different Stocks
> Our proposed framework is inherently stock-agnostic. Users of LOB Bench can evaluate their models on any asset within the LOBSTER universe – or even beyond, provided the data is converted into the LOBSTER format. For this study, we selected two stocks that differ along key metrics and are representative of assets that models could be trained on in practice.
>
> While we acknowledge that different asset classes, such as ETFs, may yield different results, exploring this was beyond the scope of our work, which focuses on presenting the benchmark itself. We also refer reviewer Nn9N to our response to reviewer KDf5 regarding “Limited Assets” and “Zero-Shot Transfer Evaluation” for additional context on our stock selection.
>
> ## Sensitivity to Different Market Conditions
> We consider the evaluation of different market conditions to be distinct from the evaluation of additional stocks. A single stock can experience varying market conditions over time, including fluctuations in volatility, spread, and trading volume. Our benchmark directly accounts for this by evaluating conditional distributions.
>
> For instance, the conditional distribution of message inter-arrival times, given the spread, captures how a model reflects variations in inter-arrival times across different market conditions (i.e., spread regimes). This approach ensures that the benchmark effectively assesses a model’s ability to adapt to dynamic market environments.
>
> ## Alternative Practical Evaluation: Downstream Tasks & Sim-to-Real Transfer
> We acknowledge the concerns regarding the reliance on distributional similarity metrics, such as the L1 norm and Wasserstein-1 distance. While these metrics are valuable for comparing real and generated data, they may not fully measure the degree of practical utility of synthetic data in all real-world applications.
>
> To address this, we propose including a brief evaluation of mid-price trend forecasting—a downstream task relevant to algorithmic trading—in the camera-ready version of the paper. Specifically, we will assess model performance on a held-out set of real data after training on three different datasets: (1) real historical data, (2) a combination of real and generated data, and (3) purely generated data. This allows assessing the sim-to-real transfer gap and thus evaluating the quality of the generated model. It is worth noting, however, that this approach has its shortcomings.
>
> The implementation will follow the model architectures proposed in Prata et al. (2024), focusing on the best-performing BINCTABL architecture, along with DeepLOB and a simple LSTM. This evaluation will serve as a complementary assessment to the distributional metrics, providing insights into how synthetic data impacts mid-price forecasting accuracy. Usefulness for predicting mid-prices might be a very discrete property of generated data, and a potential failure in this regard might prove uninformative for future model development.
>
>
> ## RL Using Generated Data
> Training reinforcement learning (RL) agents using generated data in a limit order book (LOB) setting is an active area of ongoing and future research. This approach represents another instance of sim-to-real transfer evaluation. Generative models have the potential to overcome the limitations of training policies solely on static historical data by introducing dynamic, action-dependent data trajectories, thereby enriching the training environment.
>
> ## Grammar and Spelling
> Thank you for highlighting these specific typos. We have now corrected them in the manuscript.

---

### Official Review · Reviewer_KDf5 · 2025-03-14

**Overall Recommendation:** 3

**Summary:**

This paper introduces LOB-Bench, a novel benchmark implemented in Python for evaluating the quality and realism of generative AI models applied to Limit Order Book (LOB) data in the LOBSTER format. The benchmark addresses the lack of quantitative evaluation paradigms in financial sequence modeling by providing a comprehensive framework for distributional evaluation. LOB-Bench measures distributional differences between generated and real LOB data, both conditionally and unconditionally, using a suite of relevant LOB statistics (spread, volume, imbalance, inter-arrival times) and scores from a discriminator network. Furthermore, it incorporates "market impact metrics" to assess cross-correlations and price response functions for specific events. The authors benchmark several generative models, including autoregressive state-space models, a (C)GAN, and a parametric LOB model, finding that autoregressive GenAI approaches outperform traditional models. The code and generated data are publicly available to facilitate further research and model development.

**Claims And Evidence:**

Yes, the claims made in the submission are generally well-supported by clear and convincing evidence. The paper claims to introduce a novel benchmark and demonstrate its utility in evaluating generative models for LOB data. This claim is supported by:

* Development of LOB-Bench: The paper clearly describes the components of LOB-Bench, including the various scoring functions, evaluation metrics (L1 norm, Wasserstein-1 distance), and conditional evaluation methodologies. The availability of the code further strengthens this claim.

*Benchmarking Experiments: The authors present experimental results comparing several generative models (LOBS5, baseline, Coletta, RWKV4, RWKV6) on GOOG and INTC stock data. Figures 3, 4, and 5 visually and quantitatively demonstrate the performance differences across models and metrics.

* Identification of Model Derailment: Figure 5 and related discussion provide evidence for "model derailment" in longer unrolls, a crucial insight for generative sequence models.

* Reproducibility: The paper explicitly mentions the availability of code and generated data, enhancing the reproducibility and verifiability of their findings.

The evidence presented, particularly the comparative benchmarking results and the analysis of model derailment, effectively supports the paper's claims about the value and utility of LOB-Bench.

**Essential References Not Discussed:**

Based on my familiarity with the literature on generative financial modeling and limit order book research, there do not appear to be essential related works that are critically missing from the paper's citations and discussion.

**Experimental Designs Or Analyses:**

The experimental designs and analyses are generally sound and valid for the purpose of demonstrating LOB-Bench. However, a limitation is the scope of assets considered:

* Limited Asset Universe: The experiments are primarily conducted on data for only two assets: Google (GOOG) and Intel (INTC). While these are representative stocks, evaluating the benchmark and the models on a broader range of assets, including stocks with different market capitalizations, liquidity profiles, and industry sectors, would significantly strengthen the generalizability of the findings. Restricting the analysis to just two assets limits the external validity of the conclusions about model performance and benchmark utility.

Despite this limitation, the internal validity of the experiments is well-maintained. The comparisons between models are conducted fairly using consistent evaluation metrics and the LOB-Bench framework. The analysis of model derailment and the breakdown of performance across different scoring functions are insightful and contribute to the understanding of generative LOB models.

**Methods And Evaluation Criteria:**

The proposed methods and evaluation criteria are sensible and well-justified for the problem of benchmarking generative models for LOB data. Given that this is primarily a benchmark paper, the focus is appropriately placed on the design of comprehensive and relevant evaluation criteria rather than introducing novel modeling methods.

The evaluation criteria are well-chosen because:

* Distributional Evaluation: Shifting from qualitative analysis of "stylized facts" to quantitative distributional evaluation addresses a crucial gap in the field and provides a more rigorous approach to model comparison.

* Relevant LOB Statistics: The inclusion of commonly used LOB statistics like spread, volume, imbalance, and inter-arrival times ensures that the benchmark is grounded in domain knowledge and evaluates models on features relevant to financial practitioners.

* Discriminator Network: Incorporating a discriminator network provides an adversarial perspective on model realism and captures complex, high-dimensional data characteristics that might be missed by simpler statistical metrics.

* Market Impact Metrics: Including market impact metrics assesses the model's ability to generate realistic responses to counterfactual scenarios, a key aspect for financial applications.

* Conditional Evaluation: Evaluating conditional distributions allows for a more nuanced understanding of model performance under different market conditions and over forecasting horizons, addressing the "autoregressive trap" issue.

While benchmark datasets are mentioned (LOBSTER, FI-2010), the core contribution lies in the evaluation criteria and the LOB-Bench framework itself, which are well-suited for the task of rigorously assessing generative LOB models. The focus on distributional similarity and market-relevant metrics makes the benchmark highly pertinent to the application domain.

**Other Comments Or Suggestions:**

* Expand Asset Coverage: In future work, consider benchmarking on a more diverse set of assets to demonstrate the robustness and generalizability of LOB-Bench and the evaluated models. Perhaps including stocks from different sectors, market caps, and liquidity levels.

* Explore Domain Adaptation: Briefly discuss the potential for adapting LOB-Bench or its principles to evaluate generative models in other financial domains, such as options markets, FX markets, or even broader time-series data.

* Enhance Code Documentation: While the code is available, ensure comprehensive documentation and potentially example notebooks to facilitate easier adoption and use by the community.

* Investigate "Discriminator Score Challenge" Further: Explore strategies for improving discriminator scores for generative LOB models in future research, potentially through adversarial training techniques or modified model architectures.

**Other Strengths And Weaknesses:**

Strengths:

* Originality and Significance: LOB-Bench fills a critical gap by providing the first comprehensive, quantitative benchmark for generative AI models applied to LOB data. This is a significant contribution as it enables rigorous model comparison and advancement in this important area.

* Practicality and Accessibility: The benchmark is implemented in Python and is open-source, making it easily accessible and usable by researchers and practitioners. The use of the standard LOBSTER format further enhances its practicality.

* Comprehensive Evaluation Framework: LOB-Bench incorporates a wide range of relevant metrics, including statistical distributions, discriminator scores, and market impact measures, providing a holistic view of model performance.

* Identification of Model Derailment: The paper highlights the important issue of "model derailment" in autoregressive models, providing a valuable insight for the community.

* Clarity: The paper is generally well-written and clearly explains the LOB-Bench framework, evaluation metrics, and experimental results.

Weaknesses:

* Limited Asset Scope: As mentioned before, the experiments are somewhat limited by focusing on only two assets (GOOG and INTC). Expanding the asset universe would strengthen the generalizability of the findings.

* Application Specificity: While LOB-Bench is valuable for LOB data, the paper could briefly discuss the potential for generalizing the principles of distributional benchmarking to other types of sequential financial data or time-series domains. While the framework itself is somewhat adaptable, the paper's framing is very LOB-centric.

* Discriminator Score Challenge: The paper notes that discriminator-based scoring sets a high bar. While this is a valuable metric, further discussion on how to interpret and potentially address low discriminator scores in future model development could be beneficial.

**Questions For Authors:**

* Generalizability across Assets: Given the current experiments focus on GOOG and INTC, how do you anticipate the relative performance of the benchmarked models might change when evaluated on a significantly broader and more diverse set of assets (e.g., including small-cap stocks, less liquid stocks, or stocks from different industry sectors)? Understanding the authors' perspective on asset generalizability would help assess the benchmark's broader applicability. If they anticipate significant changes in model ranking or benchmark sensitivity, it might suggest future directions for improving LOB-Bench's robustness.

* Beyond LOB Data: While LOB-Bench is specifically designed for Limit Order Book data, do you see potential avenues for adapting the core principles of distributional benchmarking and the evaluation metrics used in LOB-Bench to other types of sequential financial data or time-series forecasting tasks? Exploring the potential for broader methodological impact would increase the perceived significance of the work beyond the specific LOB domain. If authors have ideas for broader applications, it would strengthen the paper's contribution.

* Discriminator Score Interpretation: The paper notes the "difficult challenge of discriminator scores." Could you elaborate on how you interpret relatively lower discriminator scores for even state-of-the-art models? Does this primarily indicate limitations in the discriminator's ability to capture subtle differences, or are there fundamental aspects of generated LOB data that remain distinguishable from real data even for the best models? Clarification on the interpretation of discriminator scores would be valuable for guiding future model development and understanding the limitations of current generative LOB models. A nuanced response would demonstrate a deeper understanding of the evaluation methodology.

**Relation To Broader Scientific Literature:**

The key contributions of this paper are directly related to the growing body of literature on generative AI in finance, particularly in the domain of market microstructure modeling and simulation.

* Generative Financial Models: The paper builds upon recent work applying generative AI to financial data, citing papers like Nagy et al. (2023) which pioneered token-level generative modeling of LOB data. It extends this line of research by focusing on rigorous evaluation.

* LOB Simulation: The paper is directly relevant to the literature on LOB simulation, which has traditionally relied on agent-based models or parametric approaches (cited in the introduction). LOB-Bench offers a new paradigm for evaluating the realism of these simulations, particularly those powered by GenAI.

* Benchmark Datasets and Evaluation Metrics in Financial ML: The paper addresses the broader challenge of benchmarking machine learning models in finance, where evaluation has often been ad-hoc and lacking in standardized metrics. It contributes to the emerging field of financial machine learning benchmarks.

* Autoregressive Sequence Models: The paper implicitly connects to the broader literature on autoregressive sequence models, highlighting the "autoregressive trap" problem and the importance of evaluating models beyond next-token prediction accuracy, especially relevant in the context of LLMs and generative models.

While the application is tightly focused on LOB data, the methodological contribution of a comprehensive distributional benchmark for generative models is more broadly relevant to evaluating sequential generative models in other domains, even if the paper itself doesn't explicitly explore these extensions.

**Theoretical Claims:**

There are no significant theoretical claims in this paper that require formal proof verification. The paper is primarily empirical and methodological, focused on developing and demonstrating a benchmark rather than establishing new theoretical results. Therefore, this question is not applicable.

---

> ### Author Rebuttal · Authors · 2025-04-01
>
> We sincerely thank reviewer KDf5 for their detailed review and thoughtful comments, which not only provide valuable feedback but also highlight key strengths of our paper. In particular, we appreciate the recognition of the important gap our work addresses by introducing a well-founded, fully distributional evaluation of generative LOB data, including assessments of model derailment and price impact functions.  We address the queries and concerns below, and would welcome any follow-up questions or suggestions. If these have been suitably addressed, we would appreciate an increase in the review score to help build a stronger case for acceptance.
>
> ## Limited Assets
> The reviewer raises a concern regarding the limited number of assets evaluated in our study. While we acknowledge that results may vary across different stocks, we selected GOOG and INTC as representative examples because they span a broad spectrum of relevant statistics, such as volatility, relative tick size (tick size relative to stock price), and traded volume.
>
> Expanding the evaluation to additional stocks would require substantial computational resources. Training each considered model class on a new stock dataset takes multiple days on 8 L40S GPUs per stock. Moreover, as is standard practice in the domain, each model is trained on a single stock, since ample training data is usually available, and models are expected to specialize in the characteristics of individual stocks.
>
> Small-cap and less liquid stocks, while an interesting avenue for future research, present additional challenges beyond the scope of this paper. These stocks provide less training data due to lower trading activity, which complicates deep learning approaches. A promising direction for future work is multi-asset modeling, which could leverage data across multiple stocks to enhance training and capture stock-independent dynamics.
>
> It is important to emphasize that the benchmark itself is designed to allow researchers to evaluate models on any stock of their choice, and its usefulness is not limited to the two stocks presented in our paper. Our package includes efficient code that enables fitting baseline model parameters to new stock data in seconds to minutes. Furthermore, we plan to expand our research by developing new models that will naturally be evaluated on a broader set of stocks.
>
>
> ## Zero-Shot Transfer Evaluation
> An alternative evaluation could assess zero-shot model transfer—training on one stock (e.g., GOOG) and generating data for another (e.g., WMT). While this reduces computational costs, it introduces challenges.
>
> First, seeding mechanisms vary by model class: autoregressive models allow conditioning on history, but the Coletta CGAN model only seeds at the start of the day, and the baseline model does not support seeding. Second, it is unclear whether performance differences in transferred data stem from stock characteristics or model limitations. A more robust approach would involve training on multiple stocks, which is beyond the paper’s scope.
>
> That said, our benchmark supports such research directions, as it remains agnostic to the model training regime.
>
>
> ## Discriminator Scores
> We have added the following to the paper:
> > “High discriminability may result from model errors, as indicated by imperfect model scores [see …]. A distributional mismatch in a single scoring function can be sufficient to make fake data identifiable. To mitigate this issue, future research could evaluate adversarial performance by training a discriminator on perturbed data and reporting scores conditioned on the noise level, particularly as models improve on this benchmark.”
>
> Low discriminator scores suggest that even state-of-the-art models still struggle to generate perfectly indistinguishable LOB data. Future improvements may follow two paths: (1) training larger supervised models on richer datasets, akin to recent LLM advancements, or (2) explicitly minimizing discriminator scores, as in adversarial frameworks like GANs (Goodfellow et al., 2014) or GAIL (Ho et al., 2016).
>
>
> ## Beyond LOB Data
> To highlight the transferability of our methodology, we added the following to the paper:
> > “Our methodology formalizes and naturally extends common evaluation practices for synthetic one-dimensional time series, such as financial returns, which typically emphasize distributional similarity. Our framework enables a quantitative assessment of distributional properties in structured high-dimensional time series. By adapting the scoring functions, our approach could also be applied to financial transactions, payment data, streamed price quotes in forex markets, multi-asset limit order books, or decentralized crypto market protocols.”
>
>
> ## Documentation
> We provide a Jupyter notebook demonstrating LOB Bench usage and will further improve the package documentation before the paper’s potential publication.

---

### Official Review · Reviewer_EUwk · 2025-03-14

**Overall Recommendation:** 4

**Summary:**

This is a great study with multiple important contributions:
1. The paper introduces a new benchmark for evaluating limit order book (LOB) generated data,  applying aggregator functions to extract LOB-specific statistics and measuring the distribution distance between real and model generated data in both unconditional and conditional settings.
2. They also evaluate market impact response functions
3. They very well justify the distance measures used as evaluation score
4. They demonstrate nicely how their benchmark ranks various models, comparing across multiple scores and also plotting distributions showing a visual comparison.
5. They show that an autoregressive state-space model (LOBS5) outperforms traditional parametric and GAN-based methods on GOOG and INTC data.

**Claims And Evidence:**

Authors claim that their benchmark quantitatively assesses the realism of generative models for LOB data. They also claim that based on their findings, the LOBS5 model notably achieves superior performance over competing methods. They provide detailed analysis on their evaluation measures as well as detailed statistical analysis including error divergence curves, bootstrapped confidence intervals, discriminator ROC scores, etc. to support their claims.

**Essential References Not Discussed:**

Not that I am aware of.

**Experimental Designs Or Analyses:**

Their empirical results evaluate five different generative models (including traditional parametric methods, GAN-based approaches, and autoregressive models) on LOB data for Alphabet (GOOG) and Intel (INTC). Their analysis is robust: it employs multiple LOB-specific statistics, using a variety of LOB-specific metrics and visually contrasting the distributional differences in synthetic data across several tasks. It also visualizes error accumulation and histogram discrepancies over time, and uses bootstrapped confidence intervals to assess significance, which collectively underscore the models’ varying capabilities in capturing realistic market dynamics. Figures and captions are very clear and informative.

**Methods And Evaluation Criteria:**

Their proposed framework maps high-dimensional LOB data to 1D scores via aggregator functions, compares histograms of these scores using the L1 norm and Wasserstein-1 distance, for both unconditional and conditional distributions (where the inference timeframe is bounded, such as error divergence over forecast horizons).

**Other Comments Or Suggestions:**

-

**Other Strengths And Weaknesses:**

The key strength of the paper is its novel and detailed evaluation framework that provides clear, quantitative measures for generative realism in LOB data. It would be beneficial to add more ablation experiments for other choices of discriminator network architectures, and other binning strategies.

**Questions For Authors:**

1. How sensitive are your benchmark results to the choice of aggregator functions?
2. How sensitive are your divergence metrics to the binning strategy?

**Relation To Broader Scientific Literature:**

This is a strong framework which provides a full distributional evaluation tool tailored to financial data, and is accessible and easily transferable to other domains.

**Theoretical Claims:**

NA

---

> ### Author Rebuttal · Authors · 2025-04-01
>
> We sincerely thank reviewer **EUwk** for their thorough review and detailed feedback. We appreciate their recognition of the key strengths of our paper and framework, including the robust justification for our methods and scoring functions, the generality of the methodology, market impact evaluations, fully distributional scoring, and the extensive visualizations provided. We address the queries below, and would welcome any follow-up questions or suggestions. If these have been suitably addressed, we would appreciate an increase in the review score to help build a strong case for acceptance.
>
> ## Discriminator Training
> We do not make any specific claims or recommendations regarding the architecture of the discriminator. Our chosen model, combining Conv1D with attention layers, performs well and consistently distinguishes generated data from real data across all generative models, with a clear gradient in discriminator certainty across models indicating a varying degree of model veracity.
>
> Rather than focusing on a particular discriminator design, we emphasize the value of using a discriminator as a learned adversarial scoring function. This approach remains valid as long as the model is sufficiently strong to learn the discrimination task effectively. Consequently, ablations on specific model characteristics are less relevant, as various state-of-the-art models could fulfill this role.
>
> The key challenge we aim to address is improving generative model performance. In this regard, our chosen discriminator effectively exposes the gap between generated and real data while providing a meaningful learning signal for future progress.
>
> ## Sensitivity to Choice of Aggregator Function
> For a detailed analysis of how benchmark results vary with the choice of aggregator function, we refer to Figure 10 in Appendix D, which presents a bar plot comparing distributional errors across models. To ensure that outliers in scores do not disproportionately impact model rankings, we also report the median and interquartile range (IQR) of scores in the model summary plots (Figure 4).
>
> ## Sensitivity of Divergence Metrics to the Binning Strategy
> Thank you for suggesting this ablation. We are now evaluating the divergence scores with bins that are half and double their current size. We will report the sensitivity of the results based on the range of resulting scores in the camera-ready version of the paper.
>
> Since we use a dynamic regular bin size determined by the Freedman-Diaconis (FD) rule, which is designed to adapt well to the distribution of the data, we do not expect significant sensitivity to changes in bin size. Furthermore, the following theoretical convergence argument supports this choice: the FD bin size minimizes the integrated mean squared error between the histogram and the theoretical data distribution. Bin size decreases at a rate of n^{1/3}, and the expected number of observations per bin increases without bound. This suggests that, for large sample sizes, our choice of bin size will not introduce substantial inaccuracies in the divergence scores.

---

### Decision · Program_Chairs · 2025-05-01

**Decision:**

Accept (poster)

**Comment:**

Congratulations for having your paper accepted!

The reviewers have provided valuable feedback to the authors for clarifying out certain issues that will improve the manuscript in the final form.
I would advise the authors take the time till the camera ready version to incorporate the feedback.

Once again, congratulations.